# Collective epithelial migration mediated by the unbinding of hexatic defects

**Dimitrios Krommydas[1], Livio N Carenza[1,2], Luca Giomi[1]***

[1]Instituut-Lorentz, Universiteit Leiden, Leiden, Netherlands; [2]Physics Department, College of Sciences, Koç University, Istanbul, Turkiye

## eLife Assessment

This **important** theoretical study shows that active hexatic topological defects in epithelia enable collective cell flows. Within the general limitations of coarse-grained hydrodynamic models in fully capturing cell-scale behavior, the study provides **compelling** evidence supporting its conclusions. These findings will be of interest to both biophysicists studying collective cell behaviors and biologists investigating epithelial flows during development.

**\*For correspondence:**
giomi@lorentz.leidenuniv.nl

**Competing interest:** The authors declare that no competing interests exist.

**Abstract** Collective cell migration in epithelia relies on *cell intercalation*: a local remodeling of the cellular network that allows neighboring cells to swap their positions. Unlike foams and passive cellular fluid, in epithelial intercalation, these rearrangements crucially depend on activity. During these processes, the local geometry of the network and the contractile forces generated therein conspire to produce a burst of remodeling events, which collectively give rise to a vortical flow at the mesoscopic length scale. In this article, we formulate a continuum theory of the mechanism driving this process, built upon recent advances toward understanding the hexatic (i.e., sixfold ordered) structure of epithelial layers. Using a combination of active hydrodynamics and cell-resolved numerical simulations, we demonstrate that cell intercalation takes place via the unbinding of topological defects, naturally initiated by fluctuations and whose late-times dynamics is governed by the interplay between passive attractive forces and active self-propulsion. Our approach sheds light on the structure of the cellular forces driving collective migration in epithelia and provides an explanation of the observed extensile activity of in vitro epithelial layers.

## Introduction

From humble soap froths (*Weaire and Hutzler, 1999*; *Graner et al., 2008*; *Marmottant et al., 2008*) down to epithelial layers, confluent cellular fluids use intercalation to flow, even in the absence of gaps and interstitial structures. At the heart of the process is a mechanism known as *topological rearrangement process of the first kind*, or T1 for brevity, through which the vertices of a honeycomb network merge and then split, thereby leading to a remodeling of the network's topology. In epithelia, this strategy allows cells to migrate, while preserving the structural integrity of the tissue, hence its major biological functionalities. These include the barrier function, which allows epithelial layers to maintain homeostasis, ensure nutrient transport, and filter out harmful pathogens, as well as extrusion and replacement of apoptotic cells (*Irvine and Wieschaus, 1994*; *Keller et al., 2000*; *Walck-Shannon and Hardin, 2014*; *Tetley et al., 2016*; *Tetley and Mao, 2018*; *Paré and Zallen, 2020*; *Rauzi, 2020*).

While relying exclusively on local rearrangements of the cellular network, intercalation gives rise to surprisingly organized patterns, where cells are able to migrate collectively over distances orders of magnitude larger than the average cell size. In the morphogenesis of *Drosophila*, for instance, cell intercalation drives a major cellular rearrangement known as germ-band extension, in which a layer

of cells initially localized in the ventral region of the developing embryo folds over the dorsal region upon extending by approximately two and a half times its initial length. In this process, individual cells persistently move across the anterior-posterior axis more than 10 times their body length. In *Drosophila*, intercalation is also the main process driving the salivary gland tube formation, in which cells radially converge toward a central pit and eventually escape from the tangent plane of the embryo, thereby giving rise to the tubular structure (*Sanchez-Corrales et al., 2018*).

In the context of cancer progression, the role of cell intercalation has been recently debated in relation to various stages of the so-called *metastatic cascade*: that is, biomechanical pathway leading to the formation of a secondary tumor (*Cheung and Ewald, 2016*). The latter is schematically divided into three main phases: (1) detachment of cell clusters from a primary tumor and invasion of the extra-cellular matrix (ECM); (2) penetration (i.e., *intravasation*), circulation, and expulsion (i.e., *extravasation*) of the clusters in and from the blood stream; and (3) colonization of a healthy tissue and the prolifera-tion of a secondary tumor. Along the cascade, metastatic cells undergo multiple phenotypic switches, aimed at maximizing their chances of success and survival within the surrounding microenvironment. These, in turn, determine the cells' motility mode, which can vary from individual (e.g., ameboid or mesenchymal) to collective (e.g., intercalation-based or flocking guided by a small number leader cells localized at the front of the cluster) (*Friedl and Gilmour, 2009*; *Haeger et al., 2020*; *Serra-Picamal et al., 2012*; *Murugan et al., 2024*).

Yet, while being highly regulated at the biochemical level (*Cavey et al., 2008*; *Yamada et al., 2005*; *Yonemura et al., 2010*; *Buckley et al., 2014*; *Engl et al., 2014*; *Zallen and Wieschaus, 2004*; *Bertet et al., 2004*), cell intercalation cannot be separated from the mechanical forces originating it and whose nature, spatial organization, and dynamics are still largely unknown. In this article, we provide a topological insight into the mechanics of cell intercalation, by leveraging recent advances toward deciphering orientational order in epithelial layers (*Li and Ciamarra, 2018*; *Pasupalak et al., 2020*; *Durand and Heu, 2019*; *Armengol-Collado et al., 2023a*; *Armengol-Collado et al., 2023b*; *Eckert et al., 2023*; *Cislo et al., 2023*). The latter originates from the cells' anisotropic shape and results in the emergence of liquid crystal phases collectively known as *p*-atics, with $p$ an integer reflecting the symmetry of the system under rotation by $2\pi/p$. The honeycomb structure of epithelial layers, in particular, has been shown to give rise to *hexatic* order (i.e., $p = 6$) at length scales ranging from one to dozens of cells, depending on the cells' density and molecular repertoire, as well as the mechan-ical properties of the substrate (*Armengol-Collado et al., 2023a*; *Armengol-Collado et al., 2023b*; *Eckert et al., 2023*). In the language of liquid crystals, the cell-wide morphological transformations underlying T1 processes can be described in terms of topological defects known as *disclinations*: that is, point-like singularities in the otherwise regular configuration of a continuous *p*-atic order parameter – that is, $\Psi_p = \langle e^{ip\vartheta} \rangle$, with $\vartheta$ the orientation of the individual building blocks and $\langle \cdots \rangle$ the ensemble average (*Giomi et al., 2022a*; *Giomi et al., 2022b*) – around which the average cellular orientation rotates by $2\pi s$, with $s = \pm 1/p, \pm 2/p \ldots$ the winding number or 'strength' of the defect. In passive two-dimensional matter, disclinations mediate the transition from solid to liquid via a process known as Kosterlitz–Thouless–Halperin–Nelson–Young (KTHNY) melting scenario (*Kosterlitz and Thouless, 1972*; *Kosterlitz and Thouless, 1973*; *Nelson and Halperin, 1979*; *Young, 1979*; *Kosterlitz, 2016*). According to this, the hierarchical unbinding of neutral defect complexes – that is, for which $\sum_i s_i = 0$ – renders the system progressively more disordered. Our working hypothesis is that the competition between active and passive forces drives a similar unbinding mechanism in epithelial layers. In the following, we clarify the various steps and possible outcomes of this process and test this hypothesis against both hydrodynamic and cell-resolved numerical simulations.

## Results

### Cell intercalation and T1 cycle

A typical cell intercalation is illustrated in *Figure 1a* for a cluster of four cells, hereafter referred to as *primary* cell cluster. A T1 occurs when the junction that connects the internal threefold coordinated vertices shrinks until they merge into a fourfold vertex and then split once more along the orthogonal direction. Such an *internal* T1, however, leaves the number and, importantly, the positions of *external* vertices of each cell unchanged. Thus, despite this concept having received little attention in the literature (see, e.g., *Staple et al., 2010*; *Fletcher et al., 2014*; *Rauzi, 2020*; *Duclut et al., 2022*;

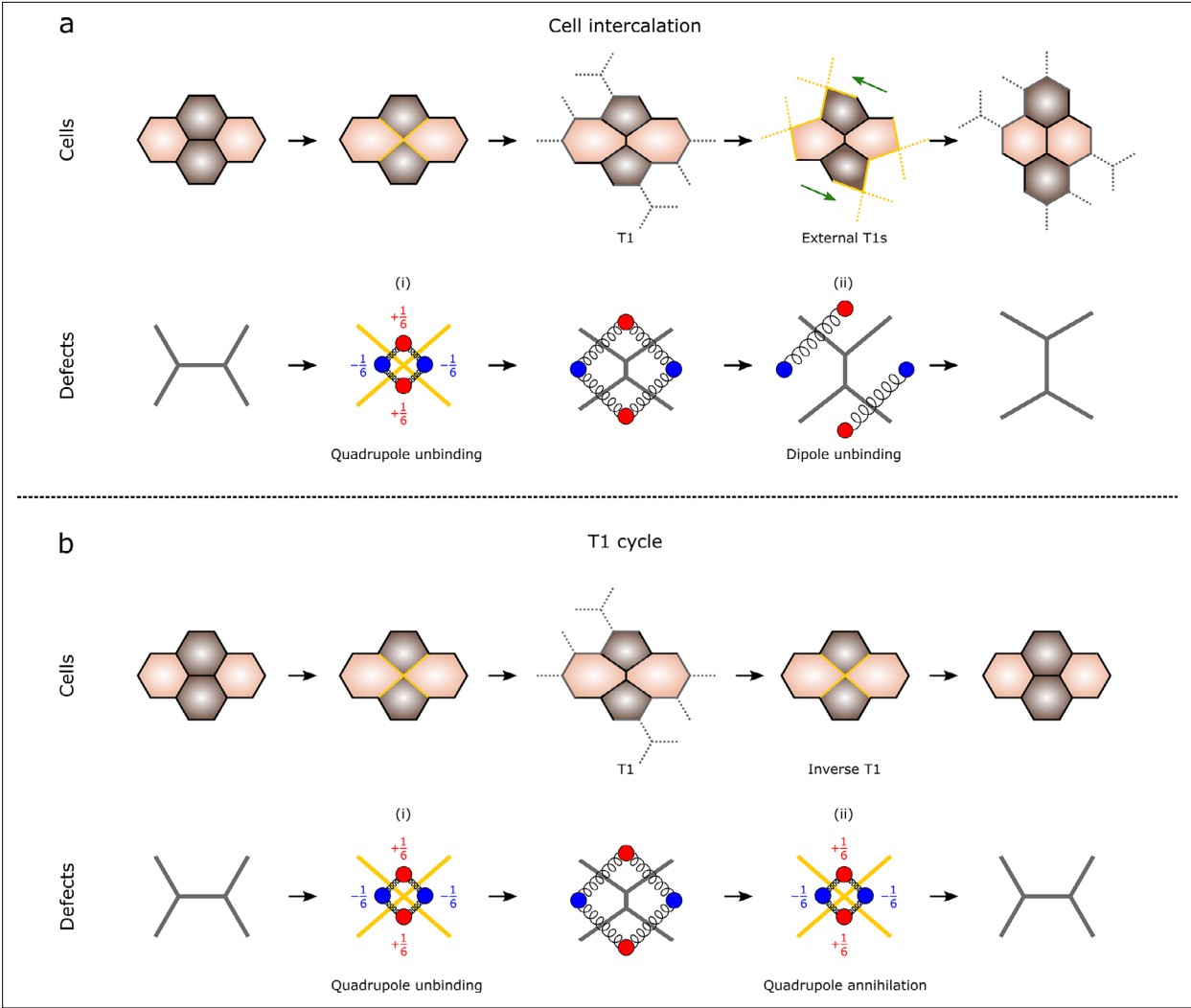

**Figure 1.** Cell intercalation and T1 cycle. (**a**) A full cell intercalation consists of an *internal* and four *external* T1 processes. The latter reconfigures the peripheral vertices of the primary cluster, thereby triggering new T1 processes across the neighboring cells. In the language of topological defects, the T1 translates to the (**i**) unbinding of a $\pm1/6$ defect quadrupole and (**ii**) a further unbinding of the quadrupole into a pair of dipoles. These two processes are schematically presented in a specific temporal order, but, in practice, they occur simultaneously or nearly so. (**b**) In a T1 cycle, the primary cell cluster undergoes a T1, followed by an inverse T1, which restores its initial configuration. The process corresponds to (**i**) the unbinding of a defect quadrupole and (**ii**) its annihilation.

*Sknepnek et al., 2023*; *Jain et al., 2023*), it is impossible to achieve collective migration by means of *isolated* T1 processes.

A full cell intercalation then consists of a burst of internal and a peripheral T1 processes, as is schematically summarized in *Figure 1a*. These steps do not necessarily occur in a sequential order but are most often simultaneous. Furthermore, since collective migration is a phenomenon that occurs roughly homogeneously across the entire cell layer, there is no single initial T1, but a uniform distribution of seeds. Crucially, internal T1 processes do *not* always trigger a full intercalation. After the first T1, where the internal threefold vertices shrink to a fourfold vertex, the cluster may reverse its dynamics and return to its original configuration (see *Figure 1b*). In the following, we refer to this scenario as the T1 *cycle*: that is, a direct T1 followed by an inverse T1, which does not permanently alter the configuration of the honeycomb network. Intuitively, and as we show next, whether the initial T1 triggers a full intercalation, hence collective migration, or a cycle, is determined by the configuration of contractile stresses exerted by the cells, which, in turn, are modulated by the local geometry of the primary cluster.

As a starting point, we focus on the intermediate configuration of primary cell cluster comprising two orthogonal pairs of pentagonal and heptagonal cells, as shown in the central column of **Figure 1**. This configuration, which corresponds to the most elementary short-ranged excitation of a honeycomb network, can be described in the language of topological defects as a quadrupole of $\pm 1/6$ disclinations. As in two-dimensional melting, such a defect structure can arise spontaneously as a consequence of spatiotemporal fluctuations of physical or biological nature. The two complementary remodeling events following the intermediate configuration, that is, cell intercalation (**Figure 1a**) and the T1 cycle (**Figure 1b**), correspond instead to the two possible 'fates' of this initial excitation. Upon unbinding (**Figure 1ai and bi**), the four defects comprising the quadrupole can further unbind into two $\pm 1/6$ disclination dipoles (**Figure 1aii**), or annihilate (**Figure 1bii**), thereby restoring the initial defect-free configuration. As we demonstrate next, the former scenario corresponds to cell intercalation and the latter to the T1 cycle. To this end, we identify three *geometric* requirements that a model of cell intercalation must fulfill, regardless of the desired level of biophysical accuracy. (1) The average orientation of the cells must rotate by $\pi/6$ with respect to initial configuration. (2) Both the primary cluster and its surrounding cells must perform a local *convergent extension*: that is, move inward in one direction, and outward along the orthogonal one (**Keller et al., 2008**; **Blanchard, 2017**; **Wang et al., 2020**; **Ioratim-Uba et al., 2023**). We stress that the adjective *local* is used here to distinguish this process from convergent extension as intended in developmental biology, where the same rearrangement occurs at the scale of the entire organism. (3) In order to remodel the external vertices and initiate cell intercalation, the primary cluster must undergo a spontaneous shear deformation.

In the following, we show that our construction not only fulfills these requirements, but, harnessing the predictive power of active hydrodynamics, provides readily testable experimental predictions. To this end, we numerically integrate the hydrodynamic equations of active hexatic liquid crystals, introduced by Armengol-Collado et al. in **Armengol-Collado et al., 2023b** (see 'Methods'). To follow the fate of the primary cell cluster after an internal T1, we assume the cells to be initially horizontally oriented, so that the phase θ of the hexatic order parameter, $\Psi_6 = |\Psi_6| e^{6i\theta}$, is $\theta = 0$, and construct a configuration featuring a quadrupole of $\pm 1/6$ disclinations (see **Figure 2ai and bi**). Along one *full* loop encircling each of these defects, θ changes by $\pm \pi/3$, with the sign reflecting that of the defect's winding number. Since the external boundary of the primary cluster consists of four *half* loops, this implies that θ varies in the range $-\pi/6 \le \theta \le \pi/6$ around the quadrupole, as indicated by the alternating blue and red tones in **Figure 2aii and bii**. Once the defect quadrupole breaks into two dipoles, this new orientation propagates from the boundary of the primary cluster into the space between the dipoles (see **Figure 2aiii**), while leaving the orientation of the cells in the exterior essentially undistorted. Thus, the unbinding of a $\pm 1/6$ defect quadrupole from a defect-free configuration and its breakup into two dipoles drives a $\pi/6$ rotation of the cells between the dipoles (see **Figure 2aiv**), consistently with our first requirement. Conversely, if defects annihilate (see **Figure 2biii**), the cells' initial orientation is restored after a transient orientational fluctuation (see **Figure 2biv**).

In order to address the second and third requirements, we look at the configuration of the velocity field $\boldsymbol{v}$, corresponding to the average velocity of the cells in the surroundings of the primary cluster. As well documented in the theoretical (**Giomi et al., 2014**; **Giomi, 2015**; **Hoffmann et al., 2020**) and experimental (**Saw et al., 2017**; **Kawaguchi et al., 2017**; **Blanch Mercader et al., 2018**; **Balasubramaniam et al., 2021**; **Yashunsky et al., 2022**) literature of active nematic liquid crystals, the distortion induced by topological defects drives a flow, whose structure and direction is determined by the defect's strength and the magnitude of the active stresses collectively exerted by the cells. Immediately after unbinding, an approximated expression for the velocity of the flow caused by the defect quadrupole can be analytically calculated (see 'Methods'). Calling $\boldsymbol{r} = |\boldsymbol{r}|(\cos\phi\,\boldsymbol{e}_x + \sin\phi\,\boldsymbol{e}_y)$ the distance from the center of the primary cluster and $\ell$ the cluster's size, that is, the distance between the center of the central junction and the center of any cell in the cluster, this velocity is given by

$$\boldsymbol{v} \approx \frac{120\alpha_6\ell^4}{\eta}\left[\left(4 - 6\log\frac{|\boldsymbol{r}|}{\ell} - 3\frac{|\boldsymbol{r}|^2}{\ell^2}\right)\cos 6\phi\right]\frac{\boldsymbol{r}}{|\boldsymbol{r}|^8}\,. \tag{1}$$

The constant $\alpha_6$ has dimensions of force over volume, embodying the active stresses exerted by the cells and modulated by the local hexatic order, and $\eta$ the shear viscosity (see 'Methods' for details). Thus, in close proximity to the primary cluster, where $|\boldsymbol{r}|/\ell \gtrsim 1$, **Equation 1** gives $v_x(x, 0) \sim \alpha_6/(\eta\ell^4)\,x$ and $v_y(y, 0) \sim -\alpha_6/(\eta\ell^4)\,y$. In agreement with our second requirement, this is a typical structure of a

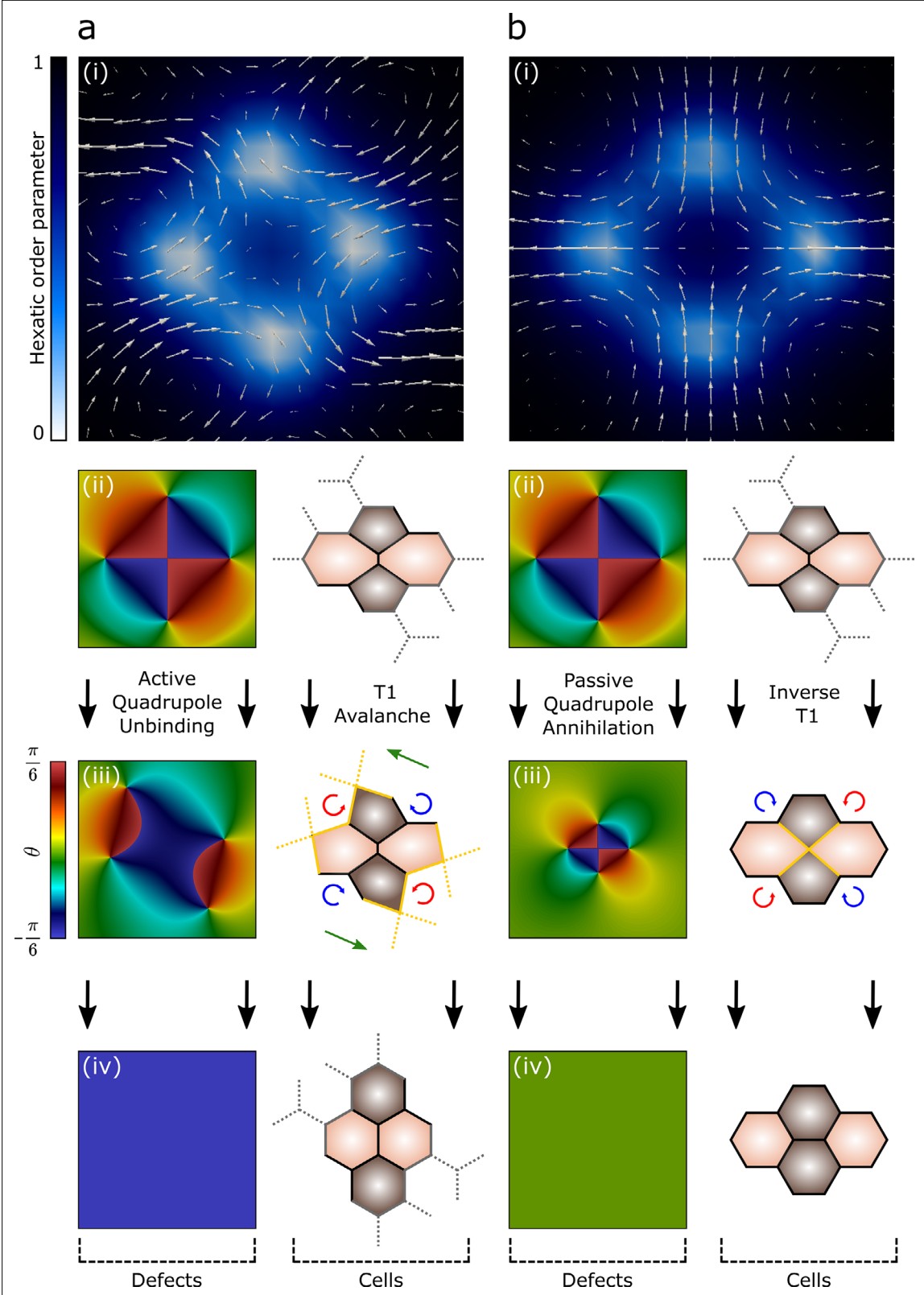

**Figure 2.** Cell intercalation and T1 cycle as defect unbinding and annihilation. (**a**) Cell intercalation. (**i**) Backflow velocity field generated during the unbinding of an *active*, hexatic defect quadrupole (*Video 1*). The three panels below show the orientation field associated with (**ii**) the quadruple in the initial configuration, (**iii**) as it unbinds in a pair of $\pm 1/6$ dipoles, and (**iv**) after the dipoles have moved outside of the region of interest, together with the corresponding configuration of the primary cluster (*Video 2*). As the dipoles move away from each other, the cells surrounding the primary cluster rotate

*Figure 2 continued on next page*

*Figure 2 continued*

clockwise (blue) and counterclockwise (red). (**b**) T1 cycle. (**i–iv**) Analogous sequence as in panel (**a**), but associated with the annihilation of the defect quadrupole. Notice that, in panel (**iii**), the direction of the flow is reversed. The details of the finite difference simulations can be found in 'Methods'.

The online version of this article includes the following source data for figure 2:

**Source data 1.** Values of the hexatic orientation field and velocity field used to produce the panels in this figure, obtained from finite-difference numerical solutions of *Equation 3a*, *Equation 3b*, *Equation 3c*.

*stagnation flow*, whose realization at the cellular scale is a local convergent extension. We stress that, consistently with the cooperative and mesoscale nature of cell intercalation, *Equation 1* cannot be obtained from the mere superposition of the flows individually sourced by the defects, as a consequence of the non-linear dependence of the order parameter on the average cellular orientation. Finally, we notice that the specific direction of motion – that is, the sign of $v$ – depends solely on $\alpha_6$, which, in turn, is either positive or negative depending on whether the active stresses exerted within the cell layer are respectively *contractile* or *extensile*.

As the cellular layer starts remodeling and the defects comprising the quadrupole migrate away from their original position, a solution of the hydrodynamic equations becomes analytically inaccessible, but can be obtained from a numerical integration of the hydrodynamic equations and is displayed in *Figure 2a and b*, for two different $\alpha_6$ values. When $\alpha_6$ is large and *negative*, the quadrupole splits into two $\pm 1/6$ dipoles moving away from each other at an angle of approximately $5\pi/6$ (see 'Methods' for details). Such a shear deformation is further enhanced by the coupling between hexatic order and flow, which, in a way not dissimilar to flow alignment effects in nematic liquid crystals, drives a rotation of the local orientation (*Gennes and Prost, 1993*). This biases the unbinding dynamics of the defect dipoles, thereby setting, in concert with the passive Coulomb-like forces at play, the direction along which the dipoles move away from each other (*Krommydas et al., 2023*). Finally, switching off active stresses – that is, $\alpha_6 = 0$, shown in *Figure 2b* – suppresses both defect unbinding and shear. Consistently, inverting the direction of active forces from extensile to contractile – that is, $\alpha_6 > 0$, see 'Methods' – results in a speed up of defects annihilation, hence of the T1 cycle, but never leads to the unbinding of $\pm 1/6$ pairs, thus to the onset of cell intercalation.

We conclude this section by stressing that the scenario emerging from our hydrodynamic analysis, of which *Equation 1* represents a central outcome, is further corroborated by recent numerical work by *Erdemci-Tandogan and Manning, 2021* and *Das et al., 2021*, who, using a different cell-resolved model of epithelia – that is, the Vertex model (*Honda and Eguchi, 1980*) – showed that the rate of T1 processes enhances the fluidity of tissues. Consistently, the speed of the stagnation flow sourced by cell intercalation increases like $\eta^{-1}$, indicating that the faster cells intercalate, the smaller $\eta$, the more fluid is the epithelial layer.

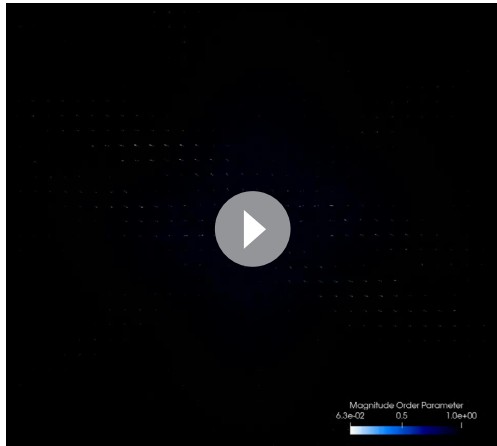

**Video 1.** Backflow velocity field (vector plot) generated during the unbinding of an active, hexatic defect quadrupole (color plot, magnitude of order parameter).
https://elifesciences.org/articles/105397/figures#video1

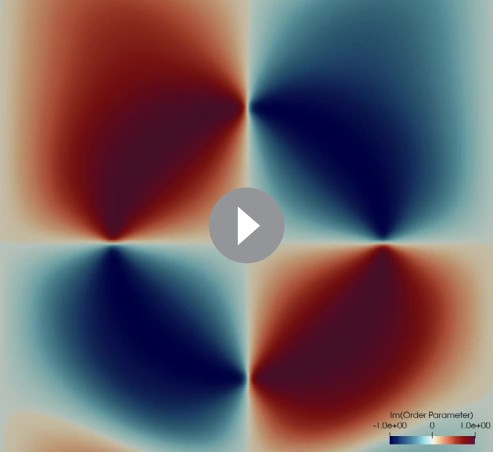

**Video 2.** Imaginaty part of hexatic order parameter (color plot) during the unbinding of an active, hexatic defect quadrupole.
https://elifesciences.org/articles/105397/figures#video2

# Collective cell migration

Having demonstrated the viability of our hydrodynamic approach, we next outline a number of general predictions, the most striking of which is that collective epithelial migration, as it results from cell intercalation, is a process of activity-guided defect unbinding. To this end, we perform numerical simulations of the multiphase field model (MPF) (*Loewe et al., 2020*), which have been proved to capture various aspects of epithelial organization, including the recently found hexanematic multiscale order (*Armengol-Collado et al., 2023a*; *Armengol-Collado et al., 2023b*; *Eckert et al., 2023*). In this approach, cells are modeled as droplets of immiscible fluid phases, undergoing a persistent random walk. Each cell is characterized by a nominal propulsion speed $v_0$, equal for all cells, and a fluctuating direction of motion, with rotational diffusion coefficient $D_r$. A detailed description of the MPF model and numerical details is provided in 'Methods'. In order to test the correlation between topological defects, T1 processes, and collective migration, we fix the speed $v_0$ and the total number of cells and vary the persistence length of the cells' trajectories by tuning the rotational diffusion coefficient $D_r$. *Figure 3a and b* show a typical configuration of MPF simulations, colored according to the normalized hexatic and nematic longitudinal stress, respectively, experienced by cells in the tissue. Note that the negative value of the longitudinal stress corresponds to extensile stresses (see the 'Methods' section for details).

To track topological defects, we first polygonize the cells by detecting their contour and marking the vertices, as shown in *Figure 3c*. For each cell, we then compute the *shape function*

$$\gamma_6 = \frac{\sum_{v=1}^{V} |\boldsymbol{r}_v|^6 e^{6i\phi_v}}{\sum_{v=1}^{V} |\boldsymbol{r}_v|^6} \ , \tag{2}$$

where $\boldsymbol{r}_v$, with $v = 1, 2 \ldots V$ is the positions of the $v$th vertex with respect to the centroid of the polygon and $\phi_v = \arctan(y_v/x_v)$ its orientation with respect to the x-axis. As demonstrated in *Armengol-Collado et al., 2023b*, the phase $Arg(\gamma_6)/6$ of the shape function identifies the sixfold orientation of a cell and is represented as white six-legged stars in *Figure 3d*. The shape function is then coarse-grained over the length scale $R = 1.5R_{\mathrm{cell}}$ to reconstruct the *shape parameter* $\Gamma_6 = \langle \gamma_6 \rangle_R$. Finally, topological defects are identified as singularities in the otherwise smoothly varying field $\Gamma_6 = \Gamma_6(\boldsymbol{r})$. T1 processes, conversely, are readily detected by tracking those events leading to a recombination of the change of neighbors of a given cell (see *Figure 3c*).

For each $D_r$ value, we reconstruct the probability of finding a T1 at a given distance from a defect (see *Figure 3c and d*) and compare it with that of an arbitrary cell. Both distributions are approximately Gaussian and have vanishing mean if expressed in terms of the signed distance $\Delta x$ (see *Figure 3e*). Furthermore, for larger $D_r$ values, where the short correlation time renders the system more disordered and dense of defects, the two probability distributions are hardly distinguishable, with the probability of finding a defect in proximity of a T1 only slightly larger than that associated with an arbitrary cell. As $D_r$ is decreased, also the density of defects decreases, and the latter probability distribution becomes flatter and flatter, while the former remains unchanged, thereby confirming that T1 processes are de facto a realization of hexatic defect unbinding. To correlate structure and dynamics, we show in *Figure 3f* the mean squared displacement of T1 cells as a function of local defect density. These measurements were taken over a time interval $\Delta t = 2.5 \times 10^4$, which aligns with the characteristic duration of T1 events in our simulations (see 'Multiphase field model of epithelial tissues' for further details). Our data confirm the existence of two different classes: that is, 'slow' and 'fast' cells, respectively denoted by purple and orange tones. While for both classes of cells, the mean squared displacement is roughly uniform for all values of the local defect density, fast cells only appear where the density is higher, thus providing an alternative signature of cell intercalation and T1 cycles. Cells undergoing a T1 cycle oscillate about their initial position but do not participate in collective migration and, therefore, exhibit small mean square displacement. By contrast, intercalating cells drive collective migration and, based on the correspondence identified here, can only be found in regions of high defect densities. Note that the difference in the mean square displacement of cells undergoing intercalation is at least one order of magnitude larger than that of cells affected by an isolated T1 cycle (*Figure 3f*). Hence, although isolated internal T1s (and by extension T1 cycles) can have small long-ranged effects, those effects are negligible in comparison to the collective long-ranged motion induced by a full cell intercalation.

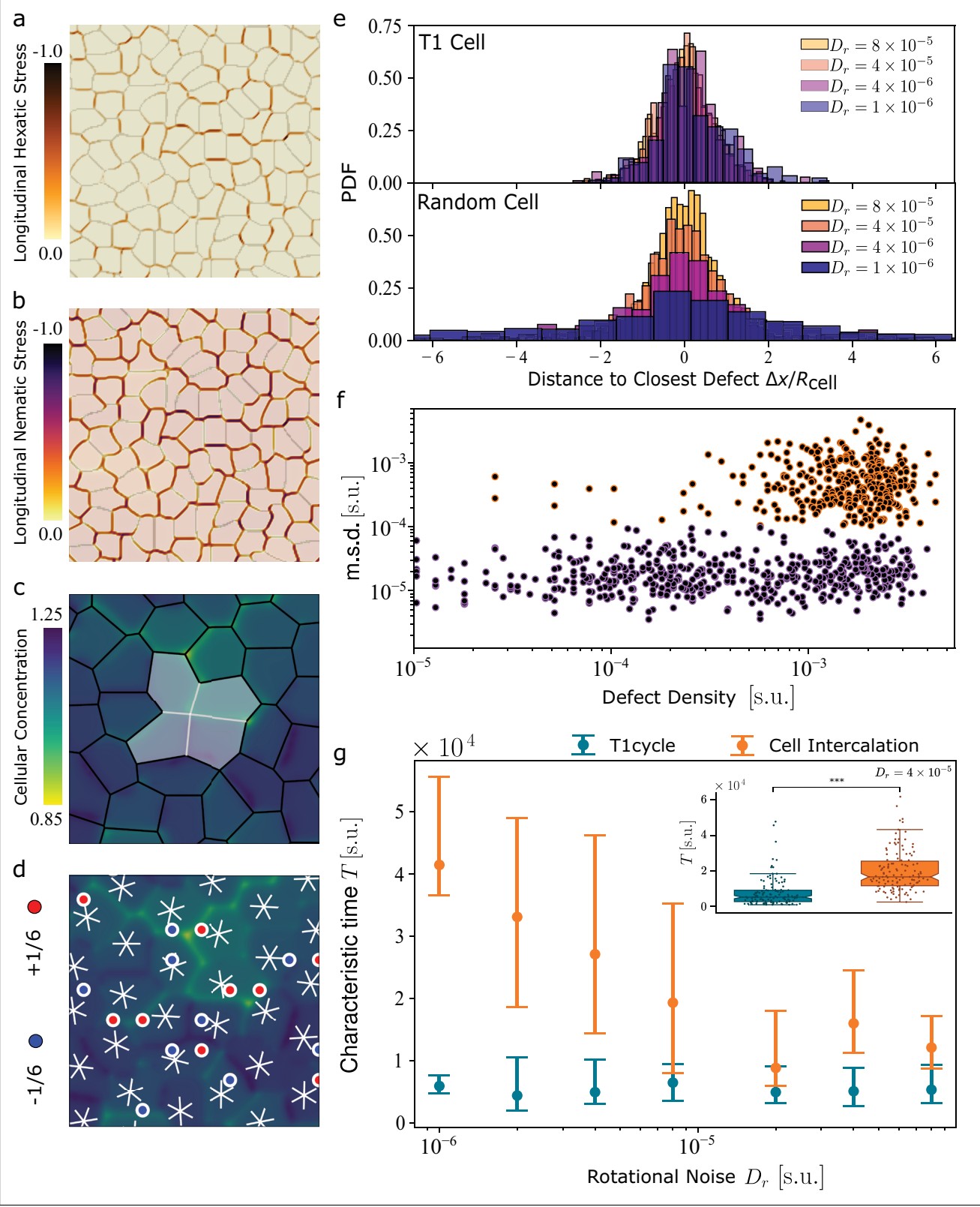

**Figure 3.** Collective cell migration as defect unbinding in the multiphase field model (MPF). (**a**, **b**) Color plots illustrating the longitudinal hexatic (**a**) and nematic (**b**) stresses in MPF simulations (refer to 'Methods'). The color bar is normalized to the largest stress magnitude observed in the configuration. Notably, the stress is uniformly negative, reflecting the extensile characteristics of both hexatic and nematic stresses. (**c**) Example of a four-cell cluster as it undergoes a T1 process, together with (**d**) the reconstructed sixfold orientation field. The six-legged stars mark the local sixfold orientation of the

*Figure 3 continued on next page*

*Figure 3 continued*

cells (see 'Methods'), while the red and blue dots denote the $+1/6$ and $-1/6$ defects. For such a four-cell cluster in real epithelial cell monolayer, please see *Armengol-Collado et al., 2023a*. (**e**) Probability distribution of finding a T1 (red tones) and a random cell (yellow tones) at a given distance from a defect, for four different values of the rotational noise $D_r$. The data indicate a prominent correlation between T1 process and topological defects. (**f**) The mean square displacement (m.s.d) of cells versus defect density computed over a time window of $\Delta t = 25 \times 10^3$ iterations, chosen to match the typical duration of T1 events and defect lifetimes. We identify two distinct subpopulations of cells: 'slow' (blue tones), with no correlation to the local density, and 'fast' (yellow tones), located where the local defect density is higher. The former corresponds to cells undergoing a T1 cycle and the latter participating in cell intercalation, hence to collective cell migration. (**g**) Temporal statistics of tissue remodeling events in multiphase field simulations. Average time between two intercalation events (orange) and average period of a T1 cycle (green) versus the rotational diffusion coefficient $D_r$. The box plot in the inset shows the statistics of events analyzed for the case at $D_r = 4 \times 10^{-5}$. (Pairwise comparisons were performed with the two-sided $t$-test: ***p $< 10^{-3}$). In the main graph, error bars are reported as the first (bottom bar) and third (upper bar) quartile of the dataset.

The online version of this article includes the following source data for figure 3:

**Source data 1.** Numerical data displayed in panels (e), (f) and (g).

Finally, following *Das et al., 2021*; *Erdemci-Tandogan and Manning, 2021* we study the temporal correlations of T1 cycles and cell intercalation. *Figure 3g* shows a plot of the average time between two intercalation events and the average period of T1 cycles, for varying rotational diffusion coefficients $D_r$. The former displays a decreasing trend with $D_r$, indicating that rotational noise improves the performance of cell intercalation, thus favoring a faster collective migration. Conversely, the dynamics of T1 cycles is essentially unaffected by rotational diffusion. Both behaviors can be readily rationalized on the basis of our hydrodynamic description. As previously observed in relation to *Figure 3e*, rotational noise favors the proliferation of hexatic defects, hence the intercalation rate. Conversely, being defect annihilation driven by the passive attractive forces resulting from the entropic elasticity of the hexatic phase, once a T1 cycle is initiated, its duration is essentially independent of the noise strength.

Before concluding, we discuss three especially striking aspects of collective cell migration highlighted by our approach and amenable to experimental scrutiny. First, from the active flow given by *Equation 1* and neglecting irrelevant inertial effects, one can estimate the magnitude of the active forces at play: that is, $F_{\text{active}} \sim \alpha_6/\ell^3$, where $\ell$ denotes the range of the distortion caused by the unbinding dislocations. Such a scaling form results primarily from the sixfold structure of cellular forces under the assumption – still to be verified in hexatic epithelial layers, but consistent with previous observations on nematic cell cultures (*Duclos et al., 2017*; *Saw et al., 2017*; *Kawaguchi et al., 2017*; *Blanch Mercader et al., 2018*; *Yashunsky et al., 2022*) – that $\alpha_6$ is, at least approximately, spatially uniform. Second, in order for these forces to prompt defect unbinding, thereby triggering cell intercalation, they must overcome the passive Coulomb-like forces driving their annihilation. At the length scale of the primary cluster, the magnitude of the latter is roughly given by $F_{\text{passive}} \sim L_6/\ell$, with $L_6$ the orientational stiffness of the hexatic phase (see 'Methods'). Equating $F_{\text{active}}$ and $F_{\text{passive}}$ provides an estimate of the typical size of the primary cluster at which the cellular layer becomes unstable to collective migration: that is, $\ell_6 = \sqrt{|\alpha_6|/L_6}$. This active hexatic length scale was identified in *Armengol-Collado et al., 2023b* and is believed to play a role analog to that of the active nematic length scale $\ell_2 = \sqrt{L_2/|\alpha_2|}$(*Giomi, 2015*). The latter is the fundamental parameter controlling the collective behavior of active nematic liquid crystals and, depending on how it compares with other extrinsic and intrinsic length scales, determines the hydrodynamic stability (*Duclos et al., 2018*) of active nematics, the distribution of the vortex area in chaotic cytoskeletal flows (*Guillamat et al., 2017*; *Lemma et al., 2019*), the size of nematic domains in bacterial colonies (*You et al., 2018*) and in vitro cultures of spindle-like cells (*Blanch Mercader et al., 2018*), etc. Similarly, we expect that confining epithelia at length scales smaller than $\ell_6$ has the effect of suppressing cell intercalation and possibly rendering necessary different locomotion strategies and possibly favoring a switch to mesenchymal phenotypes. Finally, although actomyosin networks render individual cells contractile (*Schwarz and Safran, 2002*), at a collective level, epithelial (*Saw et al., 2017*; *Blanch Mercader et al., 2018*; *Balasubramaniam et al., 2021*) and neural progenitor monolayers behave as an extensile system (*Kawaguchi et al., 2017*). In agreement with this recent experimental evidence (*Saw et al., 2017*; *Blanch Mercader et al., 2018*; *Balasubramaniam et al., 2021*), our analysis shows that only an *extensile* activity can fuel collective migration in confluent epithelia and, leveraging on the familiar language of topological defects, it further provides a simple key to rationalizing the mechanical advantage of such a biological

strategy: that is, only extensile forces can provide the type of repulsive interactions that are necessary for defects to unbind, thus to trigger cell intercalation.

## Discussion

In conclusion, we have investigated the physical mechanisms behind cell intercalation in confluent epithelial layers using analytics, and a combination of continuum and discrete modeling. After having established that cell intercalation, hence collective migration, requires a burst of correlated T1 processes, we demonstrated how the latter originates from the unbinding of neutral quadrupoles of hexatic disclinations. As in the KTHNY melting scenario (*Nelson and Halperin, 1979*), the latter are spontaneously generated by fluctuations and can either annihilate, thereby restoring the original configuration (T1 cycles), or further unbind in two pairs of $\pm 1/6$ disclinations, decreasing the translational and orientation order and increasing its fluidity. As these pairs move away from each other by activity, they stir and shear the cellular network, thereby initiating further T1 processes, which, cooperatively, give rise to cell migration. To assess the significance of our predictions, we have tested them against numerical simulations of the MPF (*Loewe et al., 2020*; *Monfared et al., 2023*; *Camley et al., 2014*; *Palmieri et al., 2015*; *Peyret et al., 2019*; *Alert and Trepat, 2020*), finding good quantitative agreement. We stress, however, that the theoretical framework employed here is not specifically tailored to the MPF or other discrete models of epithelial layers – a choice that would entail the risk of constructing a 'model of a model', rather than a model of the physical system itself. Conversely, by taking a more generic top-down approach, our theory sheds light on the structure of the cellular forces driving collective migration in epithelia, suggests the possibility of a confinement-induced switch to mesenchymal phenotypes, and provides an explanation of the observed extensile activity of in vitro epithelial layers (*Saw et al., 2017*; *Blanch Mercader et al., 2018*; *Balasubramaniam et al., 2021*).

## Methods

### Hydrodynamic equations of epithelial layers

Our hydrodynamic equations of epithelial layers have been given in *Armengol-Collado et al., 2023b* and account for both hexatic and nematic order, with the former being dominant at short and the latter at long length scales. Since cell intercalation occurs at small length scales, here we can ignore nematic order and focus solely on the hydrodynamics of the hexatic phase. In addition to the standard density $\rho$ and velocity $v$, this can be described in terms of the sixfold order parameter tensor $Q_6 = 4 \left[\!\left[ \langle \nu^{\otimes 6} \rangle \right]\!\right] = 4|\Psi_6| \left[\!\left[ n^{\otimes 6} \right]\!\right]$. In this expression, $\otimes 6$ indicates the sixfold tensorial product, $Q_6 = 4 \left[\!\left[ \langle \nu^{\otimes 6} \rangle \right]\!\right] = 4 |\Psi_6| \left[\!\left[ n^{\otimes 6} \right]\!\right]$ and $n = \cos\theta\, e_x + \sin\theta\, e_y$ are respectively the fluctuating and average orientation of the hexatic building blocks and the operator $[\![ \cdots ]\!]$ renders its argument traceless and symmetric (see *Giomi et al., 2022a*; *Giomi et al., 2022b* for a general introduction to $p$-atic hydrodynamics). The short-scale hydrodynamic equations of the epithelial layer are then given, in the most generic form, by

$$\frac{D\rho}{Dt} + \rho \nabla \cdot v = (k_{\mathrm{d}} - k_{\mathrm{a}})\rho , \tag{3a}$$

$$\rho \frac{Dv}{Dt} = \nabla \cdot \sigma + f, \tag{3b}$$

$$\frac{DQ_6}{Dt} = \Gamma_6 H_6 + 6 \left[ Q_6 \cdot \omega \right] + \lambda_6 \left[ \nabla^{\otimes 4} u \right] + \bar{\lambda}_6 \mathrm{tr}(u) Q_6 + \nu_6 \left[ u^{\otimes 3} \right]. \tag{3c}$$

with $\nu = \cos\vartheta\, e_x + \sin\vartheta\, e_y$ is the material derivative. In *Equation 3a*, $k_{\mathrm{d}}$ and $k_{\mathrm{a}}$ the cell division and apoptosis rates, here assumed equal. For simplicity, we also assume uniform density throughout the system so that *Equation 3a* reduces to the standard incompressibility condition $\nabla \cdot v = 0$. In *Equation 3b*, $\sigma$ is the total stress tensor and $[\![ \cdots ]\!]$ an external body force. In *Equation 3c*, $u = [\nabla v + (\nabla v)^{\mathsf{T}}]/2$ and $\omega = [\nabla v - (\nabla v)^{\mathsf{T}}]/2$, with $\mathsf{T}$ indicating transposition, are respectively the strain rate and vorticity tensors and entail the coupling between hexatic order and flow, with $\lambda_6$ and $\nu_6$ material constants and $\left( \nabla^{\otimes n} \right)_{i_1 i_2 \dots i_n} = \partial_{i_1} \partial_{i_2} \dots \partial_{i_n}$. Because of incompressibility, the term $\bar{\lambda}_6 \mathrm{tr}(u) Q_6$ in *Equation 3c* vanishes. The tensor $\omega = [\nabla v - (\nabla v)^{\mathsf{T}}]/2$ is the hexatic analog of the molecular tensor, dictating the relaxation dynamics of the order parameter tensor toward the minimum of the orientational free energy

$$F = \int \mathrm{d}A \left( \frac{L_6}{2} |\nabla \boldsymbol{Q}_6|^2 + \frac{A_6}{2} |\boldsymbol{Q}_6|^2 + \frac{B_6}{4} |\boldsymbol{Q}_6|^4 \right) , \tag{4}$$

where $| \cdots |^2$ is the Euclidean norm and is such that $|\boldsymbol{Q}_6|^2 = |\Psi_6|^2/2$. The constant $L_6$ is the order parameter stiffness, while $A_6$ and $B_6$ are phenomenological constants setting the magnitude of the coarse-grained complex order parameter at equilibrium: $|\Psi_6| = |\Psi_6^{(0)}| = \sqrt{-2A_6/B_6}$, when $H_{i_1 i_2 \cdots i_6} = 0$.

The stress tensor figuring in *Equation 3b* can be customarily decomposed into a passive and an active contribution: that is, $\boldsymbol{\sigma} = \boldsymbol{\sigma}^{(\mathrm{p})} + \boldsymbol{\sigma}^{(\mathrm{a})}$. The passive stress, in turn, can be expressed as $\boldsymbol{\sigma}^{(\mathrm{p})} = -P\mathbb{1} + \boldsymbol{\sigma}^{(\mathrm{v})} + \boldsymbol{\sigma}^{(\mathrm{e})} + \boldsymbol{\sigma}^{(\mathrm{r})}$, where $P$ is the pressure and $\boldsymbol{\sigma}^{(\mathrm{v})} = 2\eta \llbracket \boldsymbol{u} \rrbracket$ the viscous stress, with $\eta$ the shear viscosity. The tensor $\sigma_{ij}^{(\mathrm{e})} = -L_6 \partial_i \boldsymbol{Q}_6 \odot \partial_j \boldsymbol{Q}_6$ is the elastic stress, arising in response to a static deformation of a fluid patch and the symbol $\odot$ indicates a contraction of all matching indices of the two operands yielding a tensor whose rank equates the number of unmatched indices (two in this case). Finally $\boldsymbol{\sigma}^{(\mathrm{r})} = -\lambda_6 \nabla^{\otimes 4} \odot \boldsymbol{H}_6 + 3 \left( \boldsymbol{Q}_6 \cdot \boldsymbol{H}_6 - \boldsymbol{H}_6 \cdot \boldsymbol{Q}_6 \right)$ is the reactive stress tensor, which embodies the conservative forces arising in response to flow-induced distortions of the hexatic orientation. The active stress tensor $\boldsymbol{\sigma}^{(\mathrm{a})}$ was introduced in *Armengol-Collado et al., 2023b* on the basis of phenomenological and microscopic arguments and is given by $\boldsymbol{\sigma}^{(\mathrm{a})} = \alpha_6 \nabla^{\otimes 4} \odot \boldsymbol{Q}_6$, with $\alpha_6$ a constant.

## Multiphase field model for epithelial tissues
### The model
The multiphase field model is a cell-resolved model where each cell is described in a two-dimensional space by a concentration field $\varphi_c = \varphi_c(\boldsymbol{r})$, with $c = 1, 2 \ldots N_{\mathrm{cell}}$, and $N_{\mathrm{cell}}$ the total number of cells in the systems (*Loewe et al., 2020*; *Monfared et al., 2023*). The equilibrium state is defined by the free energy $\mathcal{F} = \int \mathrm{d}A f$, where the free energy density $f$ is

$$f = \frac{\alpha}{4} \sum_c \varphi_c^2 (\varphi_c - \varphi_0)^2 + \frac{k_\varphi}{2} \sum_c |\nabla \varphi_c|^2 + \epsilon \sum_{c<c'} \varphi_c^2 \varphi_{c'}^2 + k_{\mathrm{ad}} \sum_{c<c'} \nabla \varphi_c \cdot \nabla \varphi_{c'} + \sum_c \lambda \left( 1 - \frac{1}{\pi \varphi_0^2 R_{\mathrm{cell}}^2} \int \mathrm{d}A\, \varphi_c^2 \right)^2 . \tag{5}$$

The first two terms in the free energy represent a $\phi^4$ theory for phase separation. The parameters $\alpha > 0$ and $k_\varphi > 0$ control the nominal surface tension $\sigma = \sqrt{8 k_\varphi \alpha}$ and interfacial thickness $\xi = \sqrt{2 k_\varphi / \alpha}$, stabilizing spherical shapes in an isolated environment. This ensures the concentration field $\varphi_c$ remains close to $\varphi_0$ inside the cell and vanishes outside. The term proportional to $\epsilon > 0$ enforces volume exclusion, preventing cell overlap, while the $k_{\mathrm{ad}} > 0$ term models cell–cell adhesion, promoting tissue confluency in dense environments (*Monfared et al., 2023*). Additionally, the $\lambda > 0$ term constrains cell area near its nominal value $\pi R_{\mathrm{cell}}^2$, where $R_{\mathrm{cell}}$ is the preferred cell radius.

The phase field $\varphi_c$ evolves via the Allen–Cahn equation:

$$\partial_t \varphi_c + \boldsymbol{v}_c \cdot \nabla \varphi_c = -M \frac{\delta \mathcal{F}}{\delta \varphi_c}, \tag{6}$$

where non-equilibrium effects arise from the non-equilibrium advection term involving $\boldsymbol{v}_c = v_0 (\cos \theta_c\, \boldsymbol{e}_x + \sin \theta_c\, \boldsymbol{e}_y)$. Here, $v_0$ is the constant self-propulsion speed of the $c$th cell, and $\theta_c$ defines its migration direction. The angle $\theta_c$, in turn, follows a stochastic dynamics:

$$\frac{\mathrm{d}\theta_c}{\mathrm{d}t} = \eta_c, \tag{7}$$

with noise $\eta_c$ satisfying $\langle \eta_c(t) \eta_{c'}(t') \rangle = 2 D_r \delta_{cc'} \delta(t - t')$, where $D_r$ sets the noise diffusivity. The mobility $M$ in *Equation 8* governs the relative strength of thermodynamic relaxation versus non-equilibrium migration.

We stress that, while multiphase field models can incorporate non-equilibrium effects in various ways, here the sole source of non-equilibrium is the self-propulsion velocity $\boldsymbol{v}_c$.

Here, $\alpha > 0$ and $k_\varphi > 0$ are material parameters which can be used to tune the surface tension $\sigma = \sqrt{8 k_\varphi \alpha}$ and the interfacial thickness $\xi = \sqrt{2 k_\varphi / \alpha}$ of isolated cells and thermodynamically favor spherical cell shapes, so that the concentration field is large (i.e., $\varphi_c \approx \varphi_0$) inside the cells and zero outside. The repulsive bulk term proportional to $\epsilon > 0$ captures the fact that cells cannot overlap, while the term proportional to $k_{\mathrm{ad}} > 0$ models the interfacial adhesion between cells, ultimately favoring tissue confluency in a crowded environment (*Monfared et al., 2023*). The term proportional to $\lambda > 0$

forces the cells' area around its nominal value $\pi R_{\mathrm{cell}}^2$, with $R_{\mathrm{cell}}$ the preferential cell radius. The phase field $\varphi_c$ evolves according to the Allen–Cahn equation

$$\partial_t \varphi_c + \boldsymbol{v}_c \cdot \nabla \varphi_c = -M \frac{\delta \mathcal{F}}{\delta \varphi_c} , \tag{8}$$

where $\boldsymbol{v}_c = v_0(\cos \theta_c \, \boldsymbol{e}_x + \sin \theta_c \, \boldsymbol{e}_y)$ is the velocity at which the $c$th cell self-propels, with $v_0$ a constant speed and $\theta_c$ the angle defining the nominal direction of cell migration. The latter evolves according to the stochastic equation

$$\frac{\mathrm{d}\theta_c}{\mathrm{d}t} = \eta_c , \tag{9}$$

where $\eta_c$ is a noise term with correlation function $\langle \eta_c(t)\eta_{c'}(t') \rangle = 2D_r \delta_{cc'} \delta(t - t')$ and $D_r$ a constant controlling noise diffusivity. The constant $M$ in *Equation 8* is the mobility measuring the relevance of thermodynamic relaxation with respect to non-equilibrium cell migration. *Equation 8* is solved with a finite-difference approach through a predictor-corrector finite difference Euler scheme implementing second order stencil for space derivatives (*Carenza et al., 2019*).

We have integrated the dynamical equations with a number of cells $N_{\mathrm{cells}} = 440$ in a system of size $384 \times 403$ with periodic boundary conditions. The model parameters values used in the simulations are as follows: $\alpha = 0.2$, $k_\varphi = 0.2$, $\epsilon = 0.1$, $k_{\mathrm{ad}} = 0.005$, $\lambda = 600$, $R_{\mathrm{cell}} = 11.5$, $v_0 = 0.006$. The noise variance $D_r$ was varied in the range $10^{-6} \le D_r < 8 \times 10^{-5}$.

## Cell segmentation

In order to find the sixfold orientation of each cell, we proceeded to the cell segmentation of the simulated configuration. This procedure consists of the following steps. First, we define the thresholded density of the whole cell layer as

$$\Phi = \sum_{c=1}^{N_{\mathrm{cell}}} \vartheta_H(\varphi_c - \varphi_{\mathrm{th}}) , \tag{10}$$

where $\vartheta_H$ is the Heaviside theta function such that $\vartheta_H(x) = 1$ if $x \ge 0$ and $\vartheta_H(x) = 0$ otherwise. Here, $\varphi_{\mathrm{th}}$ is the threshold marking the boundary of each cell. In particular, we choose $\varphi_{\mathrm{th}} = \varphi_0/2$. The resulting field is $\Phi = 1$ inside each cell; $\Phi = 2$ at the interface of two cells; $\Phi = 3$ or $\Phi = 4$ on the vertices. As at the interface the field $\varphi_c$ of each cell smoothly changes from $\varphi_0$ to $0$, therefore dropping below

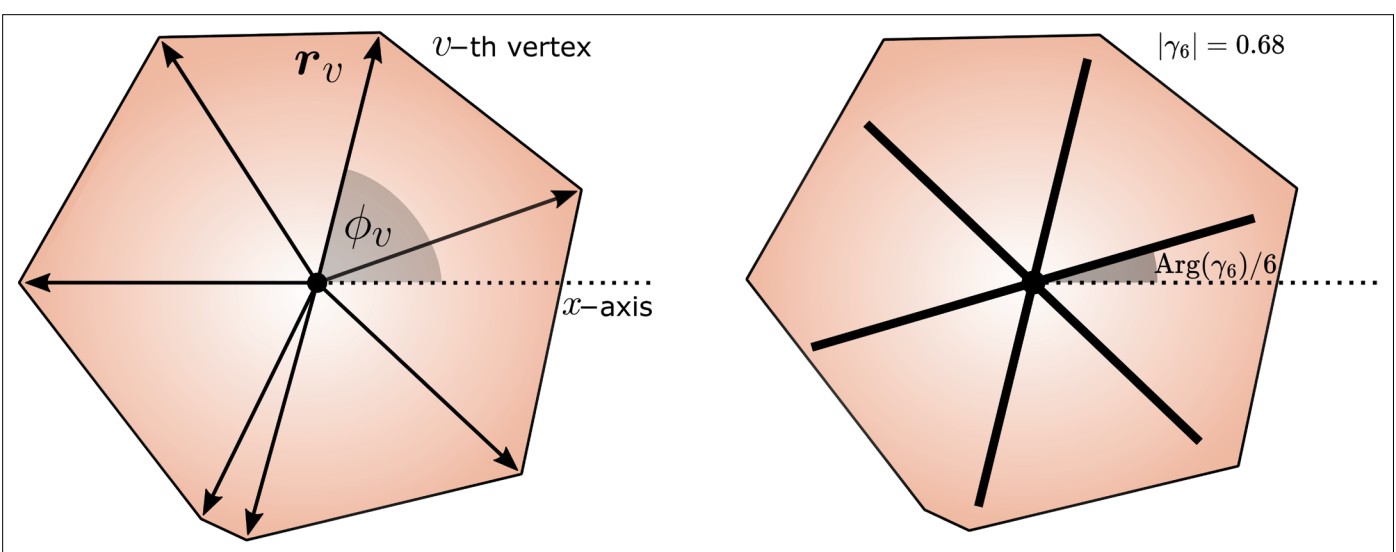

**Figure 4.** Shape function. On the left, we see a graphical representation of the sixfold shape function $\gamma_6$ (see *Equation 2* for more details) for a generic irregular polygon. On the right (black six-legged star) the phase and magnitude of $\gamma_6$ for the same cell.

the threshold $\varphi_{\mathrm{th}}$, it is possible to find pixels where $\Phi = 0$. These spurious features are adjusted by replacing **Equation 10** with an average over the pixels neighboring that where $\Phi$ vanishes. That is,

$$\Phi = \sum_{\langle x,y \rangle} \sum_{c=1}^{N_{\mathrm{cell}}} \frac{\vartheta_H(\varphi_c)}{\max\left\{\sum_{\langle x,y \rangle} \vartheta_H(\varphi_c), 1\right\}} \,, \tag{11}$$

where $\sum_{\langle x,y \rangle}$ stands for a sum over the nearest neighbors of the pixel where $\Phi = 0$. The procedure is reiterated until no change occurs in two consecutive iterations. Finally, upon segmenting the thresholded density, we identify the cell's vertices $\{r_v\}_c$, as those points where $\varphi_{\mathrm{tot}} > 2$. Tissue rearrangement events are identified by tracking changes in the list of neighbors of each cell.

## Cell orientation, coarse-graining, and topological defects

The sixfold orientation of a cell is computed via the shape function $\gamma_6$ defined in **Equation 2**, introduced in **Armengol-Collado et al., 2023a**. This construction is schematically illustrated in **Figure 4**. The single-cell orientation can then be coarse-grained to construct a continuous description of the cellular tissue (**Armengol-Collado et al., 2023a**). To do so, we use the shape order parameter $\Gamma_6 = \Gamma_6(r)$, constructed upon averaging the shape function $\gamma_6$ of the segmented cells whose center of mass, $r_c$, lies within a disk of radius $R$ centered in $r$. That is,

$$\Gamma_6(r) = \frac{1}{N_{\mathrm{disk}}} \sum_{c=1}^{N_{\mathrm{cell}}} \gamma_6(r_c)\vartheta_H(R - |r - r_c|) \,, \tag{12}$$

where $N_{\mathrm{disk}} = \sum_c \vartheta_H(R - |r - r_c|)$ is the number of cells whose centers lie within the disk, and the coarse-graining radius is fixed to be $R = 1.5 R_{\mathrm{cell}}$.

We choose to sample the shape function on a square grid with grid spacing equal to the nominal cell radius $R_{\mathrm{cell}}$. Topological defects are then identified by computing the winding number along each unit cell:

$$s = \frac{1}{2\pi} \oint_{\square} d\theta = \frac{1}{2\pi} \sum_{n=1}^{4} \left[\theta(r_{n+1}) - \theta(r_n)\right] \bmod \frac{2\pi}{6} \,, \tag{13}$$

where the symbol $\square$ denotes a square unit cell in the interpolation grid and $\theta = Arg(\Gamma_6)/6$ the phase of the shape order parameter.

## Nematic and hexatic stress in MPF simulations

The combined effect of confluency and non-equilibrium cellular migration leads to the onset of internal stresses in the tissue. These can be, in turn, classified based on their symmetry properties with respect to the cellular shape parameter. To characterize the properties of intercellular stresses in MPF simulations, we show in **Figure 3a and b** the longitudinal hexatic, nematic stress in the tissue. This is computed by contracting the nematic and hexatic concentration gradient tensor for each cell $c$ ($\sigma_c^{(\mathrm{nem})} = -k_\varphi (\nabla \phi_c)^{\otimes 2}$, $\sigma_c^{(\mathrm{hex})} = -k_\varphi (\nabla \varphi_c)^{\otimes 6}$) with the cellular shape tensor of the corresponding order ($Q_{c,p} = n_{c,p}^{\otimes p}$, with $p = 2, 6$). Here, $n_{c,p} = (\cos\theta_{c,p}, \sin\theta_{c,p})$ with $\theta_{c,p} = Arg(\gamma_{c,p})/p$ the angular orientation of the complex shape function $\gamma_p$ relative to the $c$th cell (see **Figure 4**). Therefore, the explicit expression of the longitudinal nematic and hexatic stress is given by

$$\sigma_\parallel^{(\mathrm{nem})} = \sum_{c=1}^{N_{\mathrm{cell}}} \sigma_c^{(\mathrm{nem})} : Q_{c,2} = -k_\varphi \sum_{c=1}^{N_{\mathrm{cell}}} \sum_{i,j} (\partial_i \varphi_c)(\partial_j \varphi_c)(n_{2,c})_i(n_{2,c})_j, \tag{14}$$

$$\sigma_\parallel^{(\mathrm{hex})} = \sum_{c=1}^{N_{\mathrm{cell}}} \sigma_c^{(\mathrm{hex})} : Q_{c,6} = -k_\varphi \sum_{c=1}^{N_{\mathrm{cell}}} \sum_{i1,\ldots i_6} (\partial_{i_1} \varphi_c)\ldots(\partial_{i_6} \varphi_c)(n_{6,c})_{i_1} \ldots (n_{6,c})_{i_6}. \tag{15}$$

Notice that positive (negative) values of the longitudinal stress correspond to contractile (extensile) stresses for both nematic and hexatic contributions.

## Correlating cellular mean square displacement and defect density

The scatter plot in **Figure 3f**, which correlates cellular mean square displacement with defect density, was constructed as follows. The system was divided into square subregions of size $\Delta\ell = 35$, each

containing approximately four cells. For each subregion, we analyzed a time window of $\Delta t = 25 \times 10^3$ iterations, measuring both the normalized mean square displacemenof cells (relative to the subregion area $\Delta \ell^2$) and the average defect density. The normalized displacement is calculated as m.s.d. $= \sum_{t=t^*}^{t^*+\Delta t} |\boldsymbol{r}_c(t) - \boldsymbol{r}_c(t-1)|^2/\Delta \ell^2$, where $t^*$ denotes the start time of the observation window. These measurements were collected for all subregions and time windows to generate the scatter plot.

The subregion size $\Delta \ell$ was selected to match the hexanematic crossover length scale, ensuring each area contains about four cells – the fundamental unit of tissue remodeling. This size balances the need to resolve defect-rich and defect-poor regions while avoiding excessive enlargement that would disguise local variations. Similarly, the observation window $\Delta t$ was chosen to correspond to the typical time required for a cell to traverse a subregion of size $\Delta \ell$. Shorter windows might miss remodeling events, while longer windows would average over different dynamical states.

## Analysis of characteristic time of cell intercalation and T1 cycles

To analyze the temporal statistics of remodeling events, we start differentiating these between T1 cycles and cell intercalation. T1 cycles are identified as events where cells first change their neighbors at time $t$ and then they return to their initial configuration at a later time $t + T$. Analogously, for cell intercalation events – leading to a permanent tissue remodeling – we measure the time interval $T$ between two consecutive intercalation events. The results of the analysis at varying the rotational diffusion $D_r$ are shown in *Figure 3g*. For each value of $D_r$ considered, we first identify cell intercalations and T1 cycles for each cell in the system, then we average the measured time intervals, obtaining the distribution of mean values, explicitly shown in the box plot in the inset of *Figure 3g*. The statistical analysis shows that the two populations (T1 cycles and intercalation events) are significantly different, with T1 cycles occurring faster than cell intercalation. We repeat this analysis at varying $D_r$, and we plot in *Figure 3g* the characteristic time obtained as the mean value of the cell intercalations and T1 cycles distributions. Importantly, we find that, while the time interval between cell intercalation events sensibly depends on the rotational diffusion, the typical timescale of T1 cycles does not.

## Active flow of a hexatic defect quadrupole

### Scalar order parameter

In this section, we provide a derivation of *Equation 1*. To this end, let $\Psi_6 = |\Psi_6|e^{6i\theta}$ be the hexatic complex order parameter and consider a quadrupole of $\pm 1/6$ disclinations equidistantly placed from the center of the primary cluster. The phase $\theta = \theta(\boldsymbol{r})$ is then given by the convolution of the average orientation in the surrounding of each defect, that is,

$$\theta = -\frac{1}{6}\arctan\left(\frac{y}{x-\ell}\right) - \frac{1}{6}\arctan\left(\frac{y}{x+\ell}\right) + \frac{1}{6}\arctan\left(\frac{y+\ell}{x}\right) + \frac{1}{6}\arctan\left(\frac{y-\ell}{x}\right), \quad (16)$$

where $\ell$ is distance from the center. The quadrupolar distance $\ell$ is by definition taken to be small compared to the size of the system. Thus, expanding *Equation 16* for $|\boldsymbol{r}|/\ell \gg 1$, we obtain the simpler expression

$$\theta = -\frac{2\ell^2 \sin 2\phi}{3|\boldsymbol{r}|^2} + \mathcal{O}\left(|\ell/\boldsymbol{r}|^6\right). \quad (17)$$

Notice that the Taylor expansion features *only* the quadrupolar term of order $\ell^2$ and is exact up to sixth order in $\ell/|\boldsymbol{r}|$; that is, the dipolar term, of order $\mathcal{O}(\ell/|\boldsymbol{r}|)$, and all other terms up to the sixth order vanish identically.

This result is extremely robust and can be derived in a number of ways. For instance, *Equation 16* can be obtained from the solution of the Poisson equation

$$\nabla^2 \varphi = \rho_{\mathrm{d}} \quad (18)$$

where φ is a dual field such that $\partial_i \theta = -\epsilon_{ij}\partial_j \varphi$ and the right-hand side of the *Equation 18* is analogous to the electrostatic charge density (*Chaikin et al., 1995*). At large distance from the defects, *Equation 18* can be solved by multipole expansion (*Jackson, 1999*), that is,

$$\varphi = a_0 \log \frac{r_0}{|\mathbf{r}|} + \sum_{n=1}^{\infty} \frac{a_n \cos n\theta + b_n \sin n\theta}{|\mathbf{r}|^n} \ , \tag{19}$$

where $r_0$ is an irrelevant length scale and $a_n$ and $b_n$ are coefficients given by

$$a_n = \frac{1}{n} \int dA \, |\mathbf{r}|^n \cos(n\phi)\rho_d \ , \tag{20a}$$

$$b_n = \frac{1}{n} \int dA \, |\mathbf{r}|^n \sin(n\phi)\rho_d \ . \tag{20b}$$

Thus, up to the quadrupole term, the expansion of $\varphi$ is given by

$$\varphi = a_0 \log \frac{r_0}{|\mathbf{r}|} + \frac{a_1 \cos\phi + b_1 \sin\phi}{|\mathbf{r}|} + \frac{a_2 \cos 2\phi + b_2 \sin 2\phi}{|\mathbf{r}|^2} + \cdots \tag{21}$$

As in electrostatics, the density $\rho_d$ is given by

$$\rho_d = \frac{1}{6}\left[ -\delta(\mathbf{r} - \ell\mathbf{e}_x) - \delta(\mathbf{r} + \ell\mathbf{e}_x) + \delta(\mathbf{r} - \ell\mathbf{e}_y) + \delta(\mathbf{r} + \ell\mathbf{e}_y) \right] \ , \tag{22}$$

where $\ell$ is again the distance from the center of the primary cluster. Now, because the defect quadrupole has, by construction, vanishing total strength and dipole moment, $a_0 = 0$ and $a_1 = b_1 = 0$. Of the quadrupolar terms, on the other hand, $a_2 = -\ell^2/3$ and $b_2 = 0$, thus

$$\varphi = -\frac{\ell^2 \cos 2\phi}{3|\mathbf{r}|^2} \ . \tag{23}$$

Finally, going from $\varphi$ to the original field $\theta$ one finds

$$\theta = -\frac{2\ell^2 \sin 2\phi}{3|\mathbf{r}|^2} \ , \tag{24}$$

thus confirming the expression given in *Equation 17*.

## Active force

To shed light on the structure of the cellular flow triggered by a T1 process, we solve the Stokes equation in the presence of an active force of the form $\mathbf{f}^{(a)} = \nabla \cdot \boldsymbol{\sigma}^{(a)}$, where

$$\boldsymbol{\sigma}^{(a)} = \alpha_6 \nabla^{\otimes 4} \odot \mathbf{Q}_6, \tag{25}$$

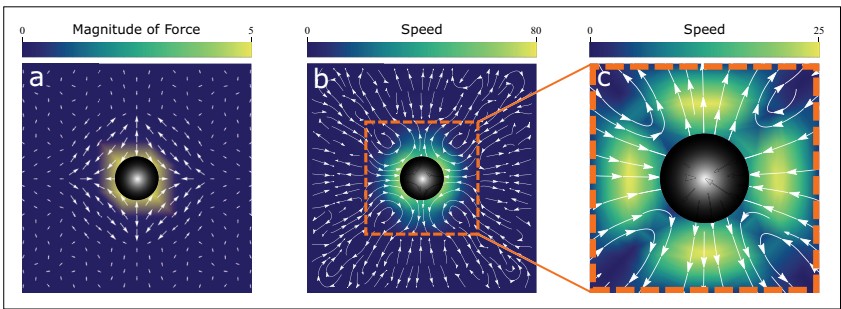

**Figure 5.** Active hexatic defect quadrupole: convergent extension analytics. (**a**) Force field: stream density plot of the force field Eq. (C11). It exhibits a clear, local, convergent-extension pattern in the vicinity of the quadrupolar radius $\ell$. (**b**) Velocity field: stream density plot of the velocity field Eq. (C17). It exhibits a clear, local, convergent-extension flow pattern in the vicinity of the quadrupolar radius $\ell$. (**c**) Velocity field approximated close to defect core: Stream density plot of the velocity field *Equation 1*. It exhibits a clear, local, convergent-extension flow pattern in the vicinity of the quadrupolar radius $\ell$. In all plots, the black disk corresponds to the radius of the quadrupole. Our analytical solution is valid outside the disk.

is the active hexatic stress tensor introduced in *Armengol-Collado et al., 2023b*. Calculating the divergence gives

$$
\begin{aligned}
\boldsymbol{f}^{(a)} &= 960\,\frac{\alpha_6 \ell^2}{|\boldsymbol{r}|^7}\left\{\left[-3\cos 7\phi + \frac{\ell^2}{|\boldsymbol{r}|^2}\left(3\cos 5\phi - 14\cos 9\phi\right)\right]\right. \\
&\left.\boldsymbol{e}_x + \left[3\sin 7\phi - \frac{\ell^2}{|\boldsymbol{r}|^2}\left(3\sin 5\phi - 14\sin 9\phi\right)\right]\boldsymbol{e}_y\right\},
\end{aligned}
\tag{26}
$$

up to correction of order $\mathcal{O}(|\ell/r|^6)$. A plot of the force field is shown in *Figure 5a*.

## Flow field

To reconstruct the cellular motion generated by a T1 process, we solve the incompressible Stokes equation for the flow sourced the active force $\boldsymbol{f}^{(a)}$: that is,

$$
\eta \nabla^2 \boldsymbol{v} - \nabla P + \boldsymbol{f}^{(a)} = \boldsymbol{0},
\tag{27a}
$$

$$
\nabla \cdot \boldsymbol{v} = 0,
\tag{27b}
$$

where $\eta$ is the shear viscosity and $P$ the pressure. To this end, we turn to the Oseen formal solution

$$
\boldsymbol{v}(\boldsymbol{r}) = \int_0^{2\pi} d\phi' \int_\ell^R dr' r'\, \boldsymbol{G}(\boldsymbol{r} - \boldsymbol{r}') \cdot \boldsymbol{f}^{(a)}(\boldsymbol{r}'),
\tag{28}
$$

where

$$
\boldsymbol{G}(\boldsymbol{r}) = \frac{1}{4\pi\eta}\left[\left(\log\frac{\mathcal{L}}{|\boldsymbol{r}|} - 1\right)\mathbb{1} + \frac{\boldsymbol{r}\otimes\boldsymbol{r}}{|\boldsymbol{r}|^2}\right],
\tag{29}
$$

is the two-dimensional Oseen tensor (see, e.g., *Giomi, 2015*), with $\mathcal{L}$ a constant, and $R$ is a large distance cut-off. Without loss of generality, one can set $\mathcal{L} = R\sqrt{e}$ in *Equation 29*. To calculate the integrals in *Equation 28*, we make use of the logarithmic expansion

$$
\log\frac{|\boldsymbol{r} - \boldsymbol{r}'|}{\mathcal{L}} = \log\frac{r_>}{\mathcal{L}} - \sum_1^\infty \frac{1}{m}\left(\frac{r_>}{r_<}\right)^m \cos\left[m(\phi - \phi')\right],
\tag{30}
$$

with $r_\gtrless$ the maximum (minimum) between $|\boldsymbol{r}|$ and $|\boldsymbol{r}'|$, and of the orthogonality of trigonometric functions

$$
\int_0^{2\pi} d\phi' \cos\left[m(\phi - \phi')\right]\cos n\phi' = \pi\cos n\phi\,\delta_{mn}.
\tag{31}
$$

The resulting flow field surrounding the defect quadrupole is then given by

$$
\begin{aligned}
\frac{\boldsymbol{v}}{2\alpha_6\ell^2/\eta} &= -\left[\frac{30\left(\ell^2 - |\boldsymbol{r}|^2\right)^2}{|\boldsymbol{r}|^{12}}\left(3|\boldsymbol{r}|^2\cos 8\phi + 14\ell^2\cos 10\phi\right) + \frac{60}{|\boldsymbol{r}|^8}\cos 6\phi\left(3|\boldsymbol{r}|^2 + 6\ell^2\log\frac{|\boldsymbol{r}|}{\ell} - 4\ell^2\right)\right]\boldsymbol{r} \\
&+ 6\left(\frac{6}{|\boldsymbol{r}|^5} - \frac{5\ell^2}{|\boldsymbol{r}|^7}\right)(\cos 5\phi\,\boldsymbol{e}_x - \sin 5\phi\,\boldsymbol{e}_y) + \frac{30}{7}\left(\frac{6\ell^2}{|\boldsymbol{r}|^7} - \frac{7}{|\boldsymbol{r}|^5}\right)(\cos 7\phi\,\boldsymbol{e}_x - \sin 7\phi\,\boldsymbol{e}_y) \\
&+ \frac{35}{3}\ell^2\left(\frac{8\ell^2}{|\boldsymbol{r}|^9} - \frac{9}{|\boldsymbol{r}|^7}\right)(\cos 9\phi\,\boldsymbol{e}_x - \sin 9\phi\,\boldsymbol{e}_y).
\end{aligned}
\tag{32}
$$

*Figure 5b* shows a plot of this flow, while *Figure 5c* shows a plot of the short distance approximation given in *Equation 1* of the main text.

## Numerical simulations of defect annihilation and unbinding

### Numerical model and validation

The time-dependent flows in *Figure 2ai* and *Figure 2bi* as well as the orientational field color maps in *Figure 2aii–iv* and *Figure 2bii–iv* are obtained by numerically integrating *Equation 3a* using a vorticity-stream function finite difference scheme. All equations are discretized on a two-dimensional square grid of sizes 256 × 256 and 1024 × 1024 with periodic boundary conditions. For both grid sizes, the grid spacing is $\Delta x = \Delta y = 1$ and the time stepping $\Delta t = 0.1$. The validity of this numerical

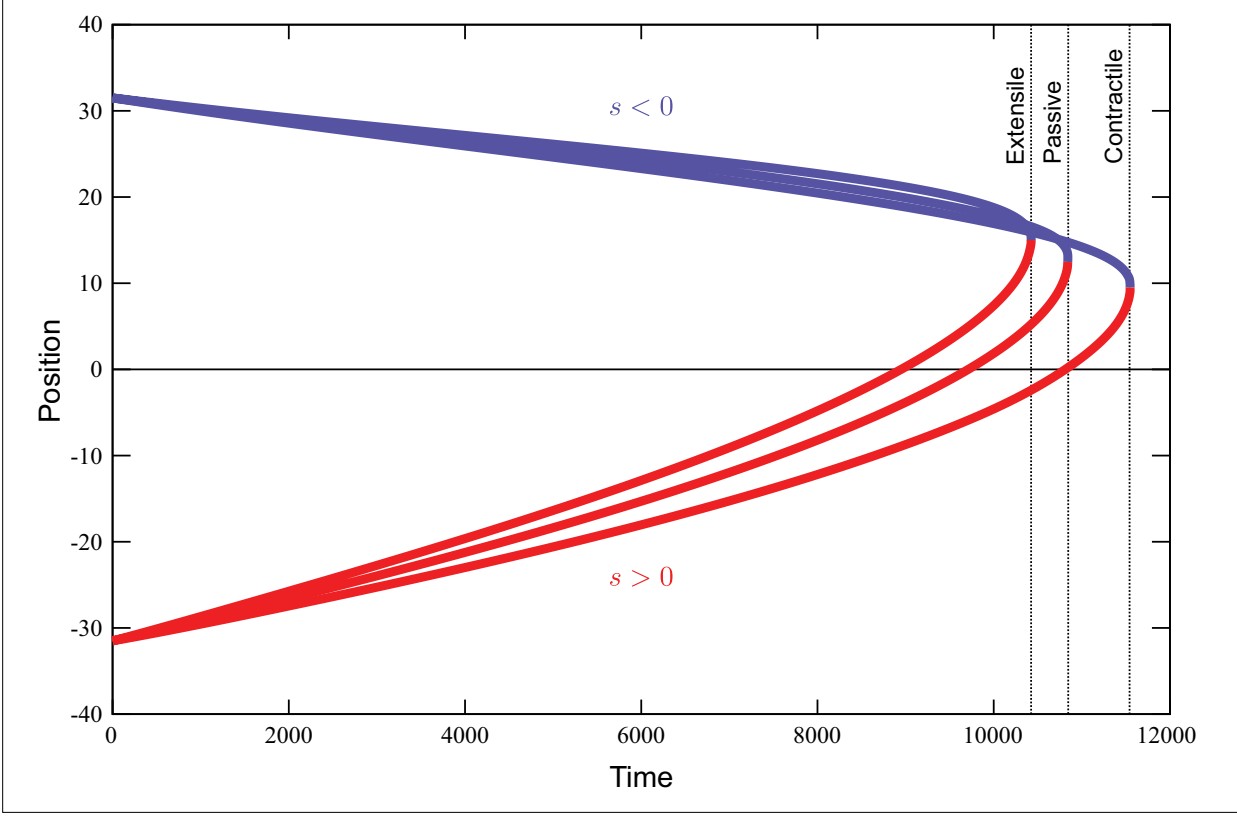

**Figure 6.** Trajectories of annihilating defects in time The red lines are the trajectories of positive and blue negative defects, respectively. Defects are sped up by positive activity ($\alpha_6 = 0.1$), but they are slowed down instead by negative activity ($\alpha_6 = -0.1$).

The online version of this article includes the following source data for figure 6:

**Source data 1.** All positions of topological defects for all iterations needed to reproduce the defect annihilation trajectory plots in this figure.

**Source data 2.** All positions of topological defects for all iterations needed to reproduce the defect annihilation trajectory plots in this figure.

**Source data 3.** All positions of topological defects for all iterations needed to reproduce the defect annihilation trajectory plots in this figure.

**Source data 4.** All positions of topological defects for all iterations needed to reproduce the defect annihilation trajectory plots in this figure.

**Source data 5.** All positions of topological defects for all iterations needed to reproduce the defect annihilation trajectory plots in this figure.

**Source data 6.** All positions of topological defects for all iterations needed to reproduce the defect annihilation trajectory plots in this figure.

**Source data 7.** All positions of topological defects for all iterations needed to reproduce the defect annihilation trajectory plots in this figure.

**Source data 8.** All positions of topological defects for all iterations needed to reproduce the defect annihilation trajectory plots in this figure.

**Source data 9.** All positions of topological defects for all iterations needed to reproduce the defect annihilation trajectory plots in this figure.

**Source data 10.** All positions of topological defects for all iterations needed to reproduce the defect annihilation trajectory plots in this figure.

**Source data 11.** All positions of topological defects for all iterations needed to reproduce the defect annihilation trajectory plots in this figure.

**Source data 12.** All positions of topological defects for all iterations needed to reproduce the defect annihilation trajectory plots in this figure.

**Source data 13.** All positions of topological defects for all iterations needed to reproduce the defect annihilation trajectory plots in this figure.

**Source data 14.** All positions of topological defects for all iterations needed to reproduce the defect annihilation trajectory plots in this figure.

**Source data 15.** All positions of topological defects for all iterations needed to reproduce the defect annihilation trajectory plots in this figure.

**Source data 16.** All positions of topological defects for all iterations needed to reproduce the defect annihilation trajectory plots in this figure.

**Source data 17.** All positions of topological defects for all iterations needed to reproduce the defect annihilation trajectory plots in this figure.

**Source data 18.** All positions of topological defects for all iterations needed to reproduce the defect annihilation trajectory plots in this figure.

approach is benchmarked by many numerical studies on liquid crystals and active matter (see for example *Krommydas et al., 2023*; *Giomi et al., 2022a*; *Giomi et al., 2022b*; *Giomi et al., 2014*). In all simulations, we set $\rho = 1$, $\eta = 1$, $L_6 = 0.5$, $A_6 = -0.2$, $B_6 = 0.4$, $\Gamma_6 = 1$ and $\lambda_6 = 1.11$. All parameters are expressed in the arbitrary units used in the numerical simulations.

## Defect annihilation and unbinding

To construct the initial configuration of $\Psi_6$, we set $\ell = 7$, $|\Psi_6| = 1$ and take $\theta$ as given in *Equation 16* inside a disk of radius $R_D = 28$ and random outside. We then thermalize this configuration by keeping the orientation of the order parameter in the disk fixed and relaxing $\Psi_6$ everywhere else. This allows us to obtain a defect-free configuration where $|\Psi_6| \approx 1$ everywhere, except that close to the defect cores where $|\Psi_6| \approx 0$. Notice that, on a doubly periodic domain, $\sum_i s_i = 0$. Therefore, no other topological defect is found at the end of such relaxation procedure. Simulations are carried out until the total free energy relative variation drops under 0.1% with respect to two consecutive iterations. This corresponds to a state where defects have annihilated, and the hexatic liquid crystal has achieved a smooth configuration everywhere in the simulation box. For both annihilation and unbinding numerical experiments, we scan $\alpha_6$ for a wide range of positive and negative values. For any negative values of activity, we obtain increasingly sheared versions of the flow pattern in *Figure 2ai*. Similarly, for positive values of $\alpha_6$ we obtain increasingly sheared versions of the same flow pattern, but with the direction of the flow inverted.

## Active defect dipole annihilation: The origin of the unbinding

In this section, we provide a brief account of the annihilation dynamics of a pair of $\pm 1/6$ active hexatic defects, in which it is possible to recognize the fundamental mechanism driving defect unbinding. To this end, we place the defects on the x-axis at a distance of $\Delta x = 64$ and construct the initial configuration of the hexatic order parameter $\Psi_6 = e^{6i\theta}$ by setting $\theta = \pm \arctan[y/(x \pm \Delta x/2)]$ inside a disk of radius $R_D = 5$ centered at the defect cores, and random outside. We thermalize this configuration by keeping the phase of the order parameter in the two disks fixed, while relaxing $\Psi_6$ everywhere else. As before, this procedure allows us to obtain a state where $|\Psi_6| \approx 1$ everywhere, except that close to the two defect cores where $|\Psi_6| \approx 0$. We use this as the initial state for our annihilation experiment. Simulations are carried out until defects have annihilated and the total free energy relative variation drops under 0.1% with respect to two consecutive iterations. The model parameters, expressed in lattice units, are again: $\Delta t = 1$, $\rho = 1$, $\eta = 1$, $L_6 = 0.5$, $A_6 = -0.2$, $B_6 = 0.4$, $\Gamma_6 = 1$, and $\lambda_6 = 1.11$.

*Figure 6* shows the trajectories of the positive (red) and negative (blue) defects during annihilation, for three realizations of the activity parameter $\alpha_6$, that is, $\alpha_6 = 0.1$ (contractile), $\alpha_6 = 0$ (passive), and $\alpha_6 = -0.1$ (extensile). For contractile activity, the backflow sourced by the active stress, *Equation 25*, annihilation is sped up with respect to the passive case. By contrast, for extensile activity, annihilation is delayed. The same effects lead to the breakup of the quadrupole into two defect pairs, provided the repulsive forces introduced by the active flow overcome the attractive Coulomb-like forces between defects.

## Acknowledgements

This work was supported by the European Union via the ERC-CoGgrant HexaTissue and by Netherlands Organization for Scientific Research (NWO/OCW). LNC acknowledges the support of the Postdoctoral EMBO Fellowship ALTF 353-2023. Part of this work was carried out on the Dutch national e-infrastructure with the support of SURF through the Grant 2021.028 for computational time. The computer codes used to produce the numerical data presented in the article are available in the manuscript.

## Additional information

### Funding

| Funder | Grant reference number | Author |
| --- | --- | --- |
| European Research Council | HexaTissue | Dimitrios Krommydas |
| European Molecular Biology Organization | ALTF 353-2023 | Livio N Carenza |

| Funder | Grant reference number | Author |
| --- | --- | --- |

The funders had no role in study design, data collection and interpretation, or the decision to submit the work for publication.

## Author contributions
Dimitrios Krommydas, Livio N Carenza, Conceptualization, Data curation, Formal analysis, Investigation, Writing – original draft, Writing – review and editing; Luca Giomi, Conceptualization, Formal analysis, Supervision, Funding acquisition, Investigation, Writing – original draft, Writing – review and editing

## Author ORCIDs
Dimitrios Krommydas ⓘD https://orcid.org/0000-0001-5936-0949
Luca Giomi ⓘD https://orcid.org/0000-0001-7740-5960

Reviewer #1 (Public review): https://doi.org/10.7554/eLife.105397.3.sa1
Reviewer #2 (Public review): https://doi.org/10.7554/eLife.105397.3.sa2
Author response https://doi.org/10.7554/eLife.105397.3.sa3

# Additional files

## Supplementary files
MDAR checklist

Source code 1. C code used for the multiphase field model simulations.

## Data availability
The current manuscript is a theoretical study. *Figure 2—source data 1*, *Figure 3—source data 1* and *Figure 6—source data 1–18* contain the numerical data used to generate the *Figures 2, 3 and 6*. The source code used to generate numerical data is enclosed as *Source code 1*.

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
