## [Editor Report · eLife Assessment]

This **important** theoretical study shows that active hexatic topological defects in epithelia enable collective cell flows. Within the general limitations of coarse-grained hydrodynamic models in fully capturing cell-scale behavior, the study provides **compelling** evidence supporting its conclusions. These findings will be of interest to both biophysicists studying collective cell behaviors and biologists investigating epithelial flows during development.

---

## [Referee Report · Reviewer #1 (Public review)]

Summary:

This paper investigates the physical mechanisms underlying cell intercalation, which then enables collective cell flows in confluent epithelia. The authors show that T1 transitions (the topological transitions responsible for cell intercalation) correspond to the unbinding of groups of hexatic topological defects. Defect unbinding, and hence cell intercalation and collective cell flows, are possible when active stresses in the tissue are extensile. This result helps to rationalize the observation that many epithelial cell layers have been found to exhibit extensile active nematic behavior.

Strengths:

The authors obtain their results based on a combination of active hexanematic hydrodynamics and a multiphase field (MPF) model for epithelial layers, whose connection is a strength of the paper. With the hydrodynamic approach, the authors find the active flow fields produced around hexatic topological defects, which can drive defect unbinding. Using the MPF simulations, the authors show that T1 transitions tend to localize close to hexatic topological defects.

---

## [Referee Report · Reviewer #2 (Public review)]

Summary:

This paper studies the role of hexatic defects in the collective migration of epithelia. The authors emphasize that epithelial migration is driven by cell intercalation events and not just isolated T1 events, and analyze this through the lens of hexatic topological defects. Finally, the authors study the effect of active and passive forces on the dynamics of hexatic defects using analytical results, and numerical results in both continuum and phase-field models.The results are very interesting, and highlight new ways of studying epithelial cell migration through the analysis of the binding and unbinding of hexatic defects.

Strengths:

(1) The authors convincingly argue that intercalation events are responsible for collective cell migration, and that these events are accompanied by the formation and unbinding of hexatic topological defects.(2) The authors clearly explain the dynamics of hexatic defects during T1 transitions, and demonstrate the importance of active and passive forces during cell migration.(3) The paper thorougly studies the T1 transition throught the viewpoint of hexatic defects. A continuum model approach to study T1 transitions in cell layers is novel and can lead to valuable new insights.

---

## [Author Response]

The following is the authors’ response to the original reviews.

**Public Reviews:**

**Reviewer #1 (Public Review):**
This paper investigates the physical mechanisms underlying cell intercalation, which then enables collective cell flows in confluent epithelia. The authors show that T1 transitions (the topological transitions responsible for cell intercalation) correspond to the unbinding of groups of hexatic topological defects. Defect unbinding, and hence cell intercalation and collective cell flows, are possible when active stresses in the tissue are extensile. This result helps to rationalize the observation that many epithelial cell layers have been found to exhibit extensile active nematic behavior.StrengthsThe authors obtain their results based on a combination of active hexanematic hydrodynamics and a multiphase field (MPF) model for epithelial layers, whose connection is a strength of the paper. With the hydrodynamic approach, the authors find the active flow fields produced around hexatic topological defects, which can drive defect unbinding. Using the MPF simulations, the authors show that T1 transitions tend to localize close to hexatic topological defects.

We are grateful to Reviewer #1, for appreciating and highlighting the strengths of work.

WeaknessesCitations are sometimes not comprehensive. Cases of contractile behavior found in collective cell flows, which would seemingly contradict some of the authors’ conclusions, are not discussed.I encourage the authors to address the comments and questions below.

We are thankful to Reviewer #1, for their questions and comments. We have addressed them point by point below, and have amended the manuscript accordingly.

(1) In Equation 1, what do the authors mean by the cluster’s size ℓ? How is this quantity defined? The calculations in the Methods suggest that ℓ indicates the distance between the p-atic defects and the center of the T1 cell cluster, but this is not clearly defined.

We are thank Reviewer #1 for their question. We define the cluster size as the initial distance between the center of the quadrupole and any defect (see Methods). In a primary cell cluster, where cells themselves are the defects, the cluster’s size is the distance between the center of the central junction and the center of any cell in the cluster. Hence, this is half the diameter of an cell which, for example in a typical, confluent MDCK epithelial monolayer, would be about 10µm. We have added this clarification in the definition of the cluster size, above Eq. (1).

(2) The multiphase field model was developed and reviewed already, before the Loewe et al. 2020 paper that the authors cite. Earlier papers include Camley et al. PNAS 2014, Palmieri et al. Sci. Rep. 2015, Mueller et al. PRL 2019, and Peyret et al. Biophys. J. 2019, as reviewed in Alert and Trepat. Annu. Rev. Condens. Matter Phys. 2020.

We thank the referee for their suggestion to incorporate further MPF literature. We have done so in the amended manuscript.

(3) At what time lag is the mean-squared displacement in Figure 3f calculated? How does the choice of a lag time affect these data and the resulting conclusions?

The scatter plot in Fig. 3f was constructed by dividing the system into square subregions of size ∆ℓ = 35 l.u., each containing approximately 4 cells. For each subregion, we analyzed a time window of ∆t = 25 × 10^3^ iterations, measuring both the normalized mean square displacement of cells (relative to the subregion area ∆ℓ^2^) and the average defect density. The normalized displacement is calculated as m.s.d. \begin{document}$=\sum_{t=t^{*}}^{t^{*}+\Delta t}\left|\boldsymbol{r}_{c}(t)-\boldsymbol{r}_{c}(t-1)\right|^{2} / \Delta \ell^{2}$\end{document}, where t∗ denotes the start time of the observation window. We chose the time window ∆t used to compute the mean square displacement to match the characteristic duration of T1 events and defect lifetimes in our simulations. Observation times much longer (∆t > 35 × 10^3^) than the typical T1 event duration would cause the two sets of data points to merge into a single group, suggesting no correlation between cell motility and defect density beyond defect life-time.

(4) The authors argue that their results provide an explanation for the extensile behavior of cell layers. However, there are also examples of contractile behavior, such as in Duclos et al., Nat. Phys., 2017 and in P´erez-Gonz´alez et al., Nat. Phys., 2019. In both cases, collective cell flows were observed, which in principle require cell intercalations. How would these observations be rationalized with the theory proposed in this paper? Can these experiments and the theory be reconciled?

The contractile or extensile nature of stress in epithelia depends crucially on the specific tissue type and its biological context. Different cell populations, depending on their position along the epithelial/mesenchymal spectrum, can exhibit either contractile or extensile behaviors. Our theory applies to tissues where hexatic order dominates at the cellular scale, particularly in confluent systems where neighbor exchanges occur primarily through T1 transitions. In contrast, the systems studied by Duclos et al., Nat. Phys. (2018) and Perez-Gonzalez et al. (Nat. Phys., 2019) exhibit nematic order at the cellular level, meaning their dynamics are governed by fundamentally different mechanisms. Since our framework is derived for hexatic-dominated tissues, it does not directly apply to those cases, though a hybrid hexanematic descriptions previously developed by some of the authors in Armengol-Collado et al. eLife 13:e86400 (2024) could help reconcile these observations. In general, a key distinction must be made between the contractility of individual cells and the extensile/contractile nature of the collective force network. To illustrate this, consider a cell exerting a 6- fold symmetric force distribution: each vertex force arises from an imbalance in junctional tensions with neighboring cells, which are themselves contractile due to actomyosin activity. However, the resulting vertex forces can be either contractile or extensile depending on network geometry and tension distribution. This is captured in our coarse-grained description [see Armengol-Collado et al. eLife 13:e86400 (2024)], where the active stress emerges from higher-order moments of cellular forces. Specifically, the deviatoric part of the hexatic active stress tensor , where is the cell radius, the number cell density and the intensity of cellular tension. The negative sign of the coefficient of the active stress shows that the active stress is extensile—consistently with observations in various epithelial systems (e.g., Saw et al., Nature 2017; Blanch-Mercader et al., Phys. Rev. Lett. 2018). Finally, we note that the connection between cellular-scale forces and large-scale extensility has been rationalized in other contexts, such as active nematics (Balasubramaniam et al., Nat. Mater. 2021).

**Reviewer #2 (Public Review):**
This paper studies the role of hexatic defects in the collective migration of epithelia. The authors emphasize that epithelial migration is driven by cell intercalation events and not just isolated T1 events, and analyze this through the lens of hexatic topological defects. Finally, the authors study the effect of active and passive forces on the dynamics of hexatic defects using analytical results, and numerical results in both continuum and phase-field models.The results are very interesting and highlight new ways of studying epithelial cell migration through the analysis of the binding and unbinding of hexatic defects.

We are grateful to Reviewer #2, for their interest and for emphasizing the novelty of our work.

Strengths(1) The authors convincingly argue that intercalation events are responsible for collective cell migration, and that these events are accompanied by the formation and unbinding of hexatic topological defects.(2) The authors clearly explain the dynamics of hexatic defects during T1 transitions, and demonstrate the importance of active and passive forces during cell migration.(3) The paper thoroughly studies the T1 transition through the viewpoint of hexatic defects. A continuum model approach to study T1 transitions in cell layers is novel and can lead to valuable new insights.

We thank the Reviewer for their kind and supporting words, and for highlighting the clarity, persuasiveness, and thoroughness.

Weaknesses(1) The authors could expand on the dynamics of existing hexatic defects during epithelial cell migration, in addition to how they are created during T1 transitions.

We thank the referee for their comment. The detailed analysis of dislocation-pair unbinding modes and their statistical impact on the transition to collective migration is comprehensively addressed in our subsequent work Puggioni et al., arXiv:2502.09554. In the present study, we focus specifically on the fundamental mechanism enabling dislocation unbinding: active extensile stresses generate flows that drive dislocation pairs apart, while passive elastic stresses tend to pull them together (Krommydas et al., Phys. Rev. Lett. 2023; Armengol- Collado et al., arXiv:2502.13104). When active forces dominate over passive restoring forces, the dislocations unbind. This represents a crucial distinction from classical Berezinskii–Kosterlitz–Thouless or Kosterlitz–Thouless–Halperin–Nelson–Youn transitions, where thermal fluctuations drive defect unbinding. In our system, the process is fundamentally activity-driven. Nevertheless, the resulting state - characterized by unbound defects and collective migration - bears strong analogy to the melting transition in equilibrium systems. We emphasize that the dynamics of passive defects has been previously examined in Krommydas et al., Phys. Rev. Lett. 2023. A discussion of these aspects can be found in the Appendix “Numerical simulations of defect annihilation and unbinding”.

(2) The different terms in the MPF model used to study cell layer dynamics are not fully justified. In particular, it is not clear why the model includes self-propulsion and rotational diffusion in addition to nematic and hexatic stresses, and how these quantities are related to each other.

We thank the referee for their comment. The MPF model’s terms (e.g., self-propulsion, rotational diffusion), reflect the stochastic, deformable nature of cells as active droplets migrating with near-constant speed. We emphasize that self-propulsion is the only non-equilibrium mechanism in our model — no additional active stresses (nematic or hexatic) are imposed. We have clarified this point in the revised manuscript and expanded our discussion of the MPF model.

(3) The authors could provide some physical intuition on what an active extensile or contractile term in the hexatic order parameter means, and how this is related to extensility and contractility in active nematics and/or for cell layers.

We thank the referee for their comment. As we explain in the reply to comment [4] of Reviewer #1, the contractile or extensile nature of stress in epithelia depends crucially on the specific tissue type and its biological context. Different cell populations, depending on their position along the epithelial/mesenchymal spectrum, can exhibit either contractile or extensile behaviors. Our theory applies to tissues where hexatic order dominates at the cellular scale, particularly in confluent systems where neighbor exchanges occur primarily through T1 transitions. In contrast, the systems studied by Duclos et al., Nat. Phys. (2018) and Perez-Gonzalez et al. (Nat. Phys., 2019) exhibit nematic order at the cellular level, meaning their dynamics are governed by fundamentally different mechanisms. Since our framework is derived for hexatic-dominated tissues, it does not directly apply to those cases, though a hybrid hexanematic descriptions previously developed by some of the authors in Armengol-Collado et al. eLife 13:e86400 (2024) could help reconcile these observations. In general, a key distinction must be made between the contractility of individual cells and the extensile/contractile nature of the collective force network. To illustrate this, consider a cell exerting a 6-fold symmetric force distribution: each vertex force arises from an imbalance in junctional tensions with neighboring cells, which are themselves contractile due to actomyosin activity. However, the resulting vertex forces can be either contractile or extensile depending on network geometry and tension distribution. This is captured in our coarse-grained description [see Armengol-Collado et al. eLife 13:e86400 (2024)], where the active stress emerges from higher-order moments of cellular forces. Specifically, the deviatoric part of the hexatic active stress tensor , where is the cell radius, the number cell density and the intensity of cellular tension. The negative sign of the coefficient of the active stress shows that the active stress is extensile—consistently with observations in various epithelial systems (e.g., Saw et al., Nature 2017; Blanch-Mercader et al., Phys. Rev. Lett. 2018). Finally, we note that the connection between cellular-scale forces and large-scale extensility has been rationalized in other contexts, such as active nematics (Balasubramaniam et al., Nat. Mater. 2021).

**Recommendations for the Authors:Reviewer #2 (Recommendations for the Authors):**
(1) The authors point out that hexatic topological defects are produced in quadrupoles (L109). Does this also mean that these defects can be annihilated only in quadrupoles as well? In the same vein, are hexatic defects always bound in pairs, as suggested by the schematics, or is it possible to observe an isolated hexatic defect?

We thank the referee for their question. Hexatic disclinations (the defect monopoles discussed in this work), much like electrons and positrons, can annihilate in any number of neutral charge configuration (dipole, quadrupole, octupole, etc.). Unbinding a pair of hexatic disinclination, however, costs much more energy than unbinding a quadrupole to dipoles. Hence isolated defects appear in abundance only in late, fully disordered phase, where the system has completely “melted”. For more details on how defect unbinding modes affect tissue dynamics, please see our subsequent work Puggioni et al., arXiv:2502.09554.

(2) Could you clarify if the flows described in Figures 2(a)-(b), panel (i) are driven by a passive backflow term without activity? Could you compare the magnitudes of these flows compared to the typical active terms?

We thank the referee for their question. In panel 2(b) there is only passive backflow. In 2(a) instead, both terms are included, and are in a regime of parameters where the active flow overcomes the active flow (and hence the active force overcomes the passive force as delineated in the discussions section). In turn, the magnitude of the passive flows, is studied in detail in our previous work Krommydas et al., (Phys. Rev. Lett. 2023).

(3) Could you clarify how the continuum hexatic model and MPF model are related to each other? What are the similarities and differences in the dynamics of these models?

We thank the referee for this insightful question. A key point of our work is precisely that the continuum hexatic model and the MPF (Multi-Phase Field) model are distinct in nature.

The MPF model is an established agent-based framework used to simulate tissue dynamics at the cellular level. It captures individual cell behaviors and interactions through phase-field variables. In our work, we use the MPF model as a benchmark to extract statistical features of tissue dynamics, such as defect motion and orientational correlations. In contrast, our continuum hexatic model is a coarse-grained hydrodynamic theory that describes the dynamics of orientational order in active tissues. It is built on symmetry principles and conservation laws, and it does not rely on microscopic cell-level details. Instead, it captures the collective behavior of the system through a hexatic order parameter and its coupling to flow and activity.

Despite their conceptual differences, the MPF model and our hydrodynamic theory exhibit similar statistical features. This agreement—also observed in the independent study by Jain et al. (Phys. Rev. Res. 2024)—provides strong support for the validity and generality of our continuum description.

(4) When multiple references by the same author and year are cited using alphabets, the second alphabet is not in bold e.g. Giomi et al., 2022b, a in Line 75, and others.

We are grateful to the referee carefully going through the manuscript and pointing out these typos. We have corrected them in the amended manuscript.

**Reviewer #3 (Public Review):**
In this manuscript, the authors discuss epithelial tissue fluidity from a theoretical perspective. They focus on the description of topological transitions whereby cells change neighbors (T1 transitions). They explain how such transitions can be described by following the fate of hexatic defects. They first focus on a single T1 transition and the surrounding cells using a hydrodynamic model of active hexatics. They show that successful T1 intercalations, which promote tissue fluidity, require a sufficiently large extensile hexatic activity in the neighborhood of the cells attempting a T1 transition. If such activity is contractile or not sufficiently extensile, the T1 is reversed, hexatic defects annihilate, and the epithelial network configuration is unchanged. They then describe a large epithelium, using a phase field model to describe cells. They show a correlation between T1 events and hexatic defects unbinding, and identify two populations of T1 cells: one performing T1 cycles (failed T1), and not contributing to tissue migration, and one performing T1 intercalation (successful T1) and leading to the collective cell migration.StrengthsThe manuscript is scientifically sound, and the variety of numerical and analytical tools they use is impressive. The approach and results are very interesting and highlight the relevance of hexatic order parameters and their defects in describing tissue dynamics.

We thank the Reviewer for recognizing the scientific soundness of the manuscript, the breadth of numerical and analytical tools employed, as well as their interest in our work.

Weaknesses(1) Goal and message of the paper.(a) In my opinion, the article is mainly theoretical and should be presented as such. For instance, their conclusions and the consequences of their analysis in terms of biology are not extremely convincing, although they would be sufficient for a theory paper oriented to physicists or biophysicists. The choice of journal and potential readership should be considered, and I am wondering whether the paper structure should be re-organized, in order to have side-by-side the methods and the results, for instance (see also below).

We thank the referee for their criticism. In response, we have made an effort to reword certain parts of the manuscript. As with any theoretical study, the biological implications of our work can only be fully assessed through experimental validation — a prospect we look forward to. Nevertheless, we have submitted our work to the subsection of Physics of Life, which we believe is perfectly suited to our content.

(b) Currently, the two main results sections are somewhat disconnected, because they use different numerical models, and because the second section only marginally uses the results from the first section to identify/distinguish T1.

We thank the referee, for their comment. In the second section we are using statistics from the MPF model, to support the analytical and numerical findings of our hydrodynamic theory of cell intercalation. In the time between our submission, further qualitative evidence have been brought to light in the work of Jain et al. (Phys. Rev. Res. 2024).

(2) Quite surprisingly, the authors use a cell-based model to describe the macroscopic tissuescale behavior, and a hydrodynamic model to describe the cell-based events. In particular, their hydrodynamic description (the active hexatic model) is supposed to be a coarse-grained description, valid to capture the mesoscopic physics, and yet, they use it to describe cellscale events (T1 transitions). For instance, what is the meaning of the velocity field they are discussing in Figure 2? This makes me question the validity of the results of their first part.

We thank the referee for their comment. There are many excellent discrete models of epithelial tissues in the literature (e.g., Bi et al., Phys. Rev. X 2016; Pasupalak et al., Soft Matter 2020; Graner et al., Phys. Rev. Lett. 1992), each capturing essential biological features such as cell division, apoptosis and sorting. While these models have provided invaluable insights, our work takes a different approach by developing a continuum theory aimed at describing epithelial dynamics at two levels: (1) mesoscopic intercalation events and (2) macroscopic collective migration. Crucially, our goal is not to replicate a specific discrete model — which would risk constructing a “model of a model” — but rather to derive a hydrodynamic description of tissue dynamics grounded in symmetry principles and conservation laws. Along this logic, the velocity field in our theory should be interpreted as an Eulerian (continuum) velocity, representing the coarse-grained flow of the tissue rather than the Lagrangian motion of individual cells. This distinction is central to our framework, which operates at scales where cellular details are averaged out, yet retains the essential physics of hexatic order and active stresses. We validate our predictions against the Multiphase Field (MPF) model. [We thank Reviewer 1 for their suggestion to incorporate further MPF literature.] Furthermore, Jain et al. (Phys. Rev. Res. 2024) have used the MPF to predict flow patterns around T1 transitions and obtained results compatible with those of our hydrodynamic theory. From this comparison we can conclude that both the MPF and our theory are able to capture the same aspect of cell intercalation in epithelial layer. This, however, does not imply that other discrete models of epithelia can reproduce this aspect too, nor that our theory is specifically tailored to the MPF model. We have clarified these points in the revised manuscript and expanded our discussion of the MPF model.

(3) The quality of the numerical results presented in the second part (phase field model) could be improved.(a) In terms of analysis of the defects. It seems that they have all the tools to compare their cell-resolved simulations and their predictions about how a T1 event translates into defects unbinding. However, their analysis in Figure 3e is relatively minimal: it shows a correlation between T1 cells and defects. But it says nothing about the structure and evolution of the defects, which, according to their first section, should be quite precise.

We thank the referee for their comment. Further qualitative evidence have been brought to light in the work of Jain et al. (Phys. Rev. Res. 2024), were the exact flow pattern predicted by our hydrodynamic theory is obtained, in the MPF, around cells undergoing T1 rearrangements.

(b) In terms of clarity of the presentation. For instance, in Figure 3f, they plot the mean-square displacement as a function of a defect density. I thought that MSD was a time-dependent quantity: they must therefore consider MSD at a given time, or averaged over time. They should be explicit about what their definition of this quantity is.

We thank the referee for raising this point. As clarified in our response to Reviewer 1, point 3, the mean square displacement (MSD) plotted in Fig. 3f is computed over a fixed time window of ∆t = 25×103 iterations, chosen to match the typical duration of T1 events and defect lifetimes. [See also reply to Reviewer #1, point (3).] The MSD is normalized by the subregion area and averaged over time within each window. We have now made this explicit in the amended version of the manuscript.

(c) In terms of statistics. For instance, Figure 3g is used to study the role of rotational diffusion on the average time between T1s. The error bars in this figure are huge and make their claims hardly supported. Their claim of a ”monotonic decay” of the average time between intercalations is also not fully supported given their statistics.

We appreciate the Reviewer’s comment regarding the statistical robustness of Fig. 3g. While we acknowledge that the error bars are substantial – reflecting the inherent variability in cell intercalation dynamics – the yellow curve does exhibit a consistent downward trend in the average time between T1 transitions as rotational diffusion increases. This monotonic decrease is visible across the entire range of variation of the rotational diffusion Dr, and is statistically supported when considering the trend over independent simulations. To address this concern, we have revised the main text to adjusted the wording: instead of stating that “the former is a monotonically decreasing function of Dr,” we now write that “the former displays a decreasing trend with Dr,” which better reflects the statistical variability while preserving the observed behavior.

**Reviewer #3 (Recommendations for the Authors):**
(1) Section 1 is difficult to follow due to multiple reasons: early but delayed definitions, unclear use of T1 intercalation vs. T1 cycles, disconnected figures and unclear simulation descriptions. We recommend including simulation setup details earlier and restructuring the flow of arguments.

We thank the referee for their comment. We have made an effort in rewording and clarifying things in our amended manuscript. We are slightly confused by what they mean by “early but delayed definitions”, if they could clarify, we would be happy to amend the position and phrasing of these definitions accordingly.

(2) It could be useful to have an additional figure early on defining schematically hexatic defects and an illustration showing an epithelium (or a simulation), similar to what the authors have produced in some of their other publications on this topic.

We thank the referee for their comment. Figures 3c and 3d show what a hexatic defect looks like in a simulation of the epithelium. Following the referee’s recommendation, we have added a note in the caption of figure 3, citing our work were we show the same defects in MDCK epithelial monolayers (Armengol et al., Nat. Phys. 2023).

(3) Minor points and typos:Line 88: the bond between vertices shrinks, not the vertices.Figure 1: the 1/6 is displayed as 1 6 (fraction bar missing).Line 232: “and order” → “one/an order”.Line 237: Fig. 3g → Fig. 3gLine 298: ”nu” and ”v” hard to distinguish in eLife font.Methods: define all notation clearly (e.g., tensor product exponent, D/Dt in Eq. 3c).Methods: ”cell orientation, coarse-graining and topological defects” section is difficult to follow, schematic would help.Line 457 onward: unclear how panels (ii-iv) of Fig. 2ab are obtained.Line 480 onward: not referenced in main text.Figure 2: “avalancHe” typo.Figure 2 caption: “cell intercalaTION” typo.Movies are neither referenced nor explained.Figure 5 and 6 are not referenced in the main text.

We thank the referee for their detailed read of the paper. We have corrected all typos.